# ITERATIVE TRAINING OF PHYSICS-INFORMED NEURAL NETWORKS WITH FOURIER-ENHANCED FEATURES

**Yulun Wu, Miguel Aguiar, Karl H. Johansson & Matthieu Barreau**
Division of Decision and Control Systems
Digital Futures and KTH Royal Institute of Technology
Stockholm, Sweden
`{yulunw,aguiar,kallej,barreau}@kth.se`

## ABSTRACT

Spectral bias, the tendency of neural networks to learn low-frequency features first, is a well-known issue with many training algorithms for physics-informed neural networks (PINNs). To overcome this issue, we propose IFeF-PINN, an algorithm for iterative training of PINNs with Fourier-enhanced features. The key idea is to enrich the latent space using high-frequency components through random Fourier features. This creates a two-stage training problem: (i) estimate a basis in the feature space, and (ii) perform regression to determine the coefficients of the enhanced basis functions. For an underlying linear model, it is shown that the latter problem is convex, and we prove that the iterative training scheme converges. Furthermore, we empirically establish that random Fourier features enhance the expressive capacity of the network, enabling accurate approximation of high-frequency PDEs. Through extensive numerical evaluation on classical benchmark problems, the superior performance of our method over state-of-the-art algorithms is shown, and the improved approximation across the frequency domain is illustrated.

## 1 INTRODUCTION

Capturing high-frequency behavior is central to modeling complex phenomena such as wave propagation, turbulence, and quantum dynamics. Traditional numerical methods, including spectral approaches (Boyd, 2001), multiscale schemes (Weinan & Engquist, 2003), and oscillatory quadrature (Iserles & Nørsett, 2005), have achieved notable success but often require problem-specific adaptations or become prohibitively costly in complex or high-dimensional settings.

There is a need for new approximation strategies that capture high-frequency behavior without sacrificing stability or tractability. Deep-learning surrogates of differential equations are a promising alternative, such as Physics-Informed Neural Networks (PINNs), which offer a grid-free alternative by combining data and physical models within a neural network framework (Raissi et al., 2017). This paradigm has shown strong performance in solving partial differential equations (PDEs) and inferring hidden dynamics, benefiting adaptability to complex geometries (Costabal et al., 2024), and high-dimensional scalability (Hu et al., 2024). Related approaches such as Fourier Neural Operators (Li et al., 2021) and DeepONet (Lu et al., 2021) further expand its reach. Despite these advances, PINN methods remain limited by *spectral bias*—the tendency of neural networks to learn low-frequency components first—which hinders accurate recovery of oscillatory solutions (Rahaman et al., 2019; Xu et al., 2025; Lin et al., 2021; Qin et al., 2024).

Several strategies have been proposed to mitigate spectral bias, including weight balancing (Wang et al., 2021a; Krishnapriyan et al., 2021), resampling (Lau et al., 2024; Tang et al., 2024; Song, 2025), and curriculum or architecture-based approaches (Sirignano & Spiliopoulos, 2018; Waheed, 2022; Chai et al., 2024; Mustajab et al., 2024; Eshkofti & Barreau, 2025; Wang & Lai, 2024). Table 1 summarizes some of the most representative approaches. While effective in certain cases, these

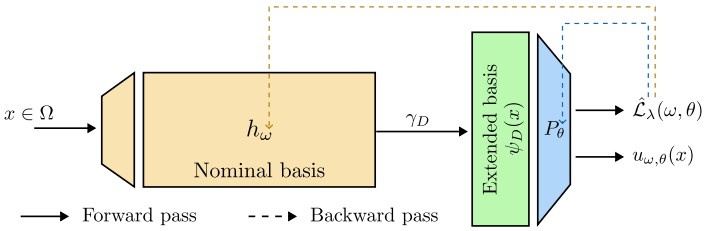

Figure 1: Architecture of IFeF-PINN. The first part (in yellow) generates the nominal basis vectors, which are then extended via $\gamma_D$ generating random Fourier features $\psi_D$ (in green), and a linear combination of the extended basis (in blue) forms the approximated solution $u_{\omega,\theta}$.

Table 1: Representative methods for approximating solutions to PDE, highlighting application domain, key idea, high-frequency handling (HF), limitations, and optimality.

| Method | Domain | Key Idea | HF | Limitations / Optimality |
|---|---|---|---|---|
| Boyd (2001); Iserles & Nørsett (2005) | Linear | Global basis functions (Fourier, Chebyshev) | +++ | Requires regular domains; **global optimum** |
| Weinan & Engquist (2003) | Multiscale | Separate scales and compute effective dynamics | ++ | Needs clear scale separation, problem-specific; **local optimum** |
| Raissi et al. (2017) | Generic | NN minimizing physics + data loss | - | Struggles with high-frequency components; **local optimum** |
| Li et al. (2021); Lu et al. (2021) | Operator | Learn mapping in Fourier / function space | + | Problem-specific, may require large networks; **local optimum** |
| Chai et al. (2024); Zhao et al. (2024) | Multiscale | Network architecture or training strategy | ++ | Problem-specific, not robust; **local optimum** |
| Lau et al. (2024); Tang et al. (2024); Song (2025) | General | Adaptive resampling | ++ | Computationally expensive, no convergence guarantees; **local optimum** |
| **IFeF-PINN** (this work) | Generic | Iterative training with extended basis via Fourier features | +++ | Not adapted to resampling, high memory footprint; **Global optimum** (for linear PDEs) |

methods remain tied to single-level optimization frameworks, where feature learning and coefficient fitting are intertwined in neural networks, limiting both robustness and theoretical guarantees.

To address this gap, we draw inspiration from classical numerical PDE solvers, which approximate solutions using basis functions, and propose a novel neural network architecture and a tailored training algorithm. The key idea is to create a feed-forward neural network with three components, as illustrated in Figure 1. First, the hidden layers $h_\omega$ generate a nominal basis in the latent functional space. Next, this basis is extended to $\psi_D$ through random Fourier features (RFF, introduced by Rahimi & Recht (2007)), which may include potentially higher-frequency elements, to span a larger latent space. Finally, the last linear layer performs regression on these extended basis vectors. The first and last blocks can be optimized separately, resulting in a two-stage iterative scheme alternating between latent basis construction and regression on output coefficients. A major feature of this framework, related to extreme learning machines (Dwivedi & Srinivasan, 2020), is that for linear differential equations, the regression stage is convex and achieves asymptotic global optimality. Unlike existing approaches, our method enriches the latent space representation, enabling systematic capture of high-frequency dynamics while leveraging the strengths of established PINN frameworks.

In this paper, we propose Iterative PINNs with Fourier-Enhanced Features (IFeF-PINN), a novel iterative two-stage training algorithm that mitigates the spectral bias of PINNs in high-frequency problems while maintaining accurate approximation on standard benchmark PDEs. Our contributions are threefold: (i) we introduce a flexible building block that augments existing PINNs architectures with improved high-frequency estimation and demonstrate its universal approximation capabilities; (ii) we propose an iterative two-stage training algorithm and prove its convergence properties; and (iii) we validate the approach through extensive simulations on benchmark problems, showing substantial improvements over existing methods.

## 2 BACKGROUND

### 2.1 PHYSICS-INFORMED NEURAL NETWORKS

PINNs is a deep learning framework that integrates PDEs into the neural network training via the loss function, enabling data-driven learning with physical constraints (Raissi et al., 2017; Karniadakis et al., 2021).

Generally, for $n > 0$, let $\Omega \subset \mathbb{R}^n$ be a bounded domain and $\mathcal{W}$ an appropriate Sobolev space of functions from $\Omega$ to $\mathbb{R}$, we consider linear PDEs of the form

$$
\begin{aligned}
\mathfrak{F}[u](x) &= f(x), & x \in \Omega, \\
\mathfrak{B}[u](s) &= g(s), & s \in \Gamma \subseteq \partial\Omega,
\end{aligned}
\tag{1}
$$

where $u \in \mathcal{W}$ is the solution, $\mathfrak{F} : \mathcal{W} \to \mathcal{L}^2(\mathbb{R}^n, \mathbb{R})$ is the linear differential operator, $f \in \mathcal{L}^2(\Omega, \mathbb{R})$ is the source term, $\mathfrak{B} : \mathcal{W} \to \mathcal{Y}(\Gamma)$ is the linear boundary/initial operator, $g \in \mathcal{Y}(\Gamma)$ specifies the boundary/initial conditions, where $\mathcal{Y}(\Gamma)$ denotes the appropriate trace space. We assume that this problem is well-posed and therefore has a unique solution in $\mathcal{W}$.

The objective of PINNs is to approximate the solution $u$ with a feedforward neural network $u_\omega$, where $\omega$ denotes the network parameters. Shin et al. (2020) and Sirignano & Spiliopoulos (2018) analyzed consistency in weak formulations under suitable assumptions, motivating the following continuum loss:

$$
\mathfrak{L}_\lambda(u_\omega) = \frac{1}{|\Gamma|} \int_\Gamma \|g(s) - \mathfrak{B}[u_\omega](s)\|^2 ds + \frac{\lambda}{|\Omega|} \int_\Omega \|\mathfrak{F}[u_\omega](x)\|^2 dx,
\tag{2}
$$

with $\lambda > 0$ where, for $A$ a bounded set, $|A|$ denotes its measure. However, this version is not numerically tractable and, in practice, we use the Monte Carlo approximation

$$
\hat{\mathfrak{L}}_\lambda(u_\omega) = \frac{1}{N_u} \sum_{i=1}^{N_u} \|g(x_u^i) - \mathfrak{B}[u_\omega](x_u^i)\|^2 + \frac{\lambda}{N_f} \sum_{i=1}^{N_f} \|\mathfrak{F}[u_\omega](x_f^i)\|^2,
\tag{3}
$$

where $\{x_u^i\}_{i=1,\dots,N_u}$ and $\{x_f^i\}_{i=1,\dots,N_f}$ are uniformly sampled on $\Gamma$ and $\Omega$, respectively. Finally, the optimal parameters are found as $\omega^* = \arg\min_\omega \hat{\mathfrak{L}}_\lambda(u_\omega)$.

### 2.2 RANDOM FOURIER FEATURES

In this work, we use random Fourier features (RFFs) introduced by Rahimi & Recht (2007) to include high-frequency terms. Grounded on Bochner's theorem, RFF provides a way to explicitly construct a feature map that approximates a stationary kernel, enabling the scaling of kernel methods to large datasets.

RFF has been used by Tancik et al. (2020) to tackle spectral bias. The novelty is to extend the input to the neural network using the RFF mapping

$$
\gamma_D(x) = \frac{1}{\sqrt{D}} \begin{bmatrix} \cos(2\pi \mathbf{B}_D x) \\ \sin(2\pi \mathbf{B}_D x) \end{bmatrix} \in \mathbb{R}^{2D},
\tag{4}
$$

where the entries of the matrix $\mathbf{B}_D \in \mathbb{R}^{D \times n}$ are sampled from a given symmetric distribution. Wang et al. (2021b) adapted this method to PINNs by using $u_\omega$ from the previous section with $2D$ inputs, so that the neural network becomes $u_\omega \circ \gamma_D$. This new architecture can learn to approximate the solution from the enriched inputs.

## 3 PROPOSED METHOD

We leverage the PINNs and RFFs in a novel way. Note first that the PINN training process couples two roles within a single nonconvex objective: (i) hidden layers $h_\omega$ learn a nonlinear feature basis, and (ii) a linear regression operator $P_\theta : h_\omega \mapsto h_\omega^\top \theta$ finds the optimal projection coefficients $\theta$ of the approximated solution onto the feature basis, thereby minimizing the loss $\hat{\mathfrak{L}}_\lambda$. This coupling leads

to PINN pathologies, where gradients from interior residuals can dominate and suppress boundary terms, and spectral bias drives low-frequency learning first, leaving oscillatory components underfit and slowing convergence on high-frequency modes (Wang et al., 2021b; 2022).

To overcome this coupling issue, we approximate the solution $u$ to the PDEs in (1) as a linear combination of basis functions. We thus consider the two problems in isolation: basis generation, which we will denote as the upper-level problem, and linear regression on the basis functions, which we will refer to as the lower-level problem.

## 3.1 THE UPPER-LEVEL PROBLEM: BASIS FUNCTION GENERATION

The initial step for the basis generation is to follow the classical PINN methodology and train a standard feed-forward neural network with parameters $(\omega, W)$, denoted by

$$\tilde{u}_{\omega,W}(x) = W h_\omega(x), \qquad x \in \Omega,$$

to minimize $\omega, W \mapsto \hat{\mathfrak{L}}_\lambda(\tilde{u}_{\omega,W})$. This is typically accomplished using a gradient-descent numerical scheme, such as ADAM (Kingma & Ba, 2014), or a more complex second-order solver, like L-BFGS (Liu & Nocedal, 1989). Then, the neural network $h_\omega : \mathbb{R}^n \to \mathbb{R}^p$ generates a basis $h_\omega \in \mathcal{C}(\mathbb{R}, \mathbb{R}^p)$ of the latent space while $W$ is the projection operator. This initial step serves as a warm-up for the upper-level problem. Note that $\tilde{u}_{\omega,W}$ most likely contains only the low-frequency components of the original solution. Therefore, the surrogate $\tilde{u}_{\omega,W}$ might be an aliased or steady-state solution of the PDE, and the fit at the boundary points might be poor.

In our approach, the strategy is to apply an RFF mapping to the last hidden layer features $h_\omega$. This upgrades the implicit linear kernel on $h_\omega$ to a stationary kernel, such as a radial basis function, in the adaptive feature space. Since $\tilde{u}$ is probably a distorted version of the real solution $u$, the RFF extension might bring higher frequency signals that mitigate the spectral bias.

Concretely, we define $\psi_D(x) = \gamma_D\left(h_\omega(x)\right) = \frac{1}{\sqrt{D}} \begin{bmatrix} \cos(2\pi \mathbf{B}_D h_\omega(x)) \\ \sin(2\pi \mathbf{B}_D h_\omega(x)) \end{bmatrix}$ where $\mathbf{B}_D \in \mathbb{R}^{D \times p}$ is a constant matrix with entries sampled i.i.d. from $\mathcal{N}(0, \sigma^2)$.

## 3.2 THE LOWER-LEVEL PROBLEM: LINEAR REGRESSION

The linear output layer over $h_\omega$ induces a dot-product kernel in feature space, which can limit expressivity and exacerbate spectral bias toward low frequencies. Applying RFF to $h_\omega$ equips the adaptive features with a stationary kernel without adding trainable parameters, injecting high-frequency components via random projections. Formally speaking, an approximate solution to the PDE in (1) with $\theta \in \mathbb{R}^{2D}$ becomes

$$u_{\omega,\theta}(x) = \psi_D(x)^\top \theta, \quad x \in \Omega. \tag{5}$$

As we show in Appendix B, since the operators $\mathfrak{F}$ and $\mathfrak{B}$ are linear, the loss function $\hat{\mathfrak{L}}_\lambda(u_{\omega,\theta})$ is quadratic in $\theta$:

$$\mathfrak{L}_{\text{lower}}(\theta \mid \omega) := \hat{\mathfrak{L}}_\lambda(u_{\omega,\theta}) = \tfrac{1}{2}\theta^\top Q(\omega)\theta + c(\omega)^\top \theta + b, \tag{6}$$

where $Q$ and $c$ collect boundary and interior residual terms.

**Proposition 1.** *Assume that $\lambda > 0$ and that the rank condition (3) from Appendix B.1 is verified. Then $Q$ is positive definite and there is a unique solution to $\arg\min_\theta \mathfrak{L}_{\text{lower}}(\theta \mid \omega) = -Q^{-1}(\omega)c(\omega)$.*

The proof is given in Appendix B.1. The application of the RFF mapping in the last hidden layer enables the generation of an arbitrary number of basis functions $\psi_D$ independently of the network's width on which we can leverage quadratic programming to get the unique optimal solution. This would otherwise not be possible because constrained by the basis dimension.

## 3.3 THE GLOBAL BI-LEVEL PROBLEM

Combining the results from the two previous subsections, we get the following formulation that decouples basis learning (upper-level) from linear regression (lower-level):

$$\omega^\star(\theta) = \arg\min_\omega \hat{\mathfrak{L}}_\lambda(u_{\omega,\theta}) := \arg\min_\omega \mathfrak{L}_{\text{upper}}(\omega \mid \theta),$$

$$\theta^\star(\omega) = \arg\min_\theta \hat{\mathfrak{L}}_\lambda(u_{\omega,\theta}) := \arg\min_\theta \mathfrak{L}_{\text{lower}}(\theta \mid \omega). \tag{7}$$

The classical bi-level optimization framework (Bard, 1991) proposes the following three-step numerical method: (i) sample $w_0, \theta_0$ randomly; (ii) solve the upper-level problem $\omega^+ = \omega^\star(\theta_0)$; (iii) solve the lower-level problem $\theta^+ = \theta^\star(\omega^+)$. The final parameters $(\omega^+, \theta^+)$ are the optimal solutions to the bi-level optimization.

However, this approach does not consider a warm start and is not particularly adapted to a learning problem. For better approximation capabilities, we propose an iterative scheme. We warm start using a vanilla PINN pre-training to get an initial value $\omega_0$ for the weights of the basis generator. Then we compute $\theta_{i+1} = \theta^*(w_i)$ before performing a one-step gradient-descent on $\omega_i$ to minimize $\mathfrak{L}_{\text{upper}}(\omega_i \mid \theta_{i+1})$ to get $\omega_{i+1}$. This leads to Algorithm 1. The convergence of this numerical scheme and the approximation capabilities of the new neural network architecture are studied in the next section.

---

**Algorithm 1** IFeF-PINN for linear PDEs

**Initialize** network parameter $w_0, \theta_0$ and $B$
**for** $k$ from 0 **to** $N_{epoch}$ **do**
    Formulate extended RFF basis $\psi_D$
    **Lower update:** $\theta_{k+1} = -Q(\omega_k)^{-1} c(\omega_k)$

    **Upper update:**
$$\omega_{k+1} = \omega_k - \eta \nabla_\omega \mathfrak{L}_{\text{upper}}(\omega_k \mid \theta_{k+1})$$

**end for**
**return** $\omega_{N_{epoch}}, \theta^\star(\omega_{N_{epoch}})$

---

*Remark* 1 (Relation to deep kernel learning). In deep kernel learning, we use a neural network to learn a nonlinear feature transformation, and a Gaussian process is defined over the resulting feature space using a traditional kernel function. This enables learning a flexible, data-driven kernel that combines the expressiveness of deep learning with the uncertainty estimation of Gaussian processes (Wilson et al., 2016). However, to the best of the authors' knowledge, learning a Gaussian process with a nonlinear PDE prior is not yet possible (Jidling et al., 2017); we propose a solution in this case.

*Remark* 2 (On the warm start). Pre-training a standard PINN for several hundred epochs provides initial network parameters for basis generation. This is necessary for homogeneous PDEs to prevent convergence to $u \equiv 0$, since standard initialization yields near-zero outputs that trivially minimize the lower-level problem. For non-homogeneous PDEs, the source term prevents this issue.

## 3.4 EXTENSION TO NONLINEAR PDEs

For nonlinear PDEs, the physics residual term $\frac{\lambda}{N_f} \sum_{i=1}^{N_f} \|\mathfrak{F}[u_{\omega,\theta}](x_f^i)\|^2$ becomes nonlinear in $\theta$, making the lower-level problem $\mathfrak{L}_{\text{lower}}(\theta \mid \omega)$ nonconvex and lacking a closed-form solution. We therefore replace the exact solution in Proposition 1 with gradient descent to find an approximate local minimizer when the Second-Order Sufficient Condition (SOSC) holds, i.e., when the gradient vanishes and the Hessian is positive definite. The complete update is given in Algorithm 2. For computational efficiency, we update $\theta$ to a local minimizer every $N_{\text{lower}}$ epochs. For initialization, we can either warm start only the network parameters $\omega$ via standard PINN pre-training as in the linear case, or initialize both $\omega$ and $\theta$ jointly via end-to-end training as discussed in Section 6.4.1.

---

**Algorithm 2** IFeF-PINN for nonlinear PDEs

**Initialize** network parameter $w_0, \theta_0$ and $B$
**for** $k$ from 0 **to** $N_{epoch}$ **do**
    Formulate extended RFF basis $\psi_D$
    **Lower update:**
    **if** $k \bmod N_{lower} = 0$ **then**
        $\theta_{k+1} \approx \arg\min_\theta \mathfrak{L}_{\text{lower}}(\omega_k \mid \theta_k)$
    **else**
        $\theta_{k+1} = \theta_k$
    **end if**
    **Upper update:**
$$\omega_{k+1} = \omega_k - \eta_\omega \nabla_\omega \mathfrak{L}_{\text{upper}}(\omega_k \mid \theta_{k+1})$$

**end for**
**return** $\omega_{N_{epoch}}, \theta^\star(\omega_{N_{epoch}})$

---

# 4 THEORETICAL ANALYSIS

## 4.1 CONVERGENCE PROPERTIES OF THE BI-LEVEL ALGORITHM

We establish convergence by showing that the optimal lower-level solution $\theta^\star(\omega)$ is Lipschitz continuous with respect to the upper-level parameters $\omega$, which ensures a well-defined Lipschitz hypergradient for gradient descent on the upper level.

**Proposition 2** (Lipschitz Continuity of the Solution Map). *Let the lower-level problem be a strongly convex QP problem parameterized by $\omega$. Assume that the mappings $\omega \mapsto Q(\omega)$ and $\omega \mapsto c(\omega)$ are locally Lipschitz continuous, and that the smallest eigenvalue of $Q(\omega)$ is uniformly bounded below*

*by $\mu_Q > 0$ on any compact set of $\omega$. Then, the optimal solution map $\theta^\star(\omega)$ is also locally Lipschitz continuous with respect to $\omega$.*

The detailed proof is provided in Appendix C.2.This also holds in the nonlinear PDE cases, when the SOSC is satisfied, the local minimizer $\theta^\star(\omega)$ retains Lipschitz continuity and differentiability in a neighborhood of $\omega$. Consequently, the hypergradient is L-smooth, which we leverage in our convergence analysis.

**Theorem 1** (Convergence to a stationary point). *Assume that 1) the functions $Q$ and $c$ are continuously differentiable with respect to $\omega$, the upper-level loss $\mathfrak{L}_{\mathrm{upper}}$ is continuously differentiable with respect to both $\theta$ and $\omega$; 2) The lower-level problem is $\mu$-strongly convex; 3) the objective function $\mathfrak{L}_{\mathrm{upper}}(\cdot \mid \theta)$ is bounded below and its hypergradient is L-smooth.*

*Then, the sequence of iterates $\{\omega_k\}_{k=0}^\infty$ generated by the gradient descent algorithm with a constant step size $\eta \in (0, 2/L)$ converges to a stationary point of $\mathfrak{L}_{\mathrm{upper}}(\cdot \mid \theta)$.*

The assumptions made are classical in learning problems and are a direct consequence of the structure of the bi-level framework. A formula for the hypergradient is derived via the Implicit Function Theorem in Appendix C.1, showing it as a composition of smooth functions. Its Lipschitz continuity is then guaranteed by the Lipschitz continuity of the solution map $\theta^\star$ established in Proposition 2.

## 4.2 UNIVERSAL APPROXIMATION CAPABILITIES

To analyze the expressiveness of the RFF-augmented features, we show that the hypothesis class is not less expressive than linear readouts over the last hidden layer features. The necessary function spaces for this analysis are defined with comprehensive foundational definitions and proofs in Appendix D.

**Definition 1.** *The **feature space** $\mathcal{H}_f$ and the **composite RFF function space** $\mathcal{H}_{\mathrm{RFF}}$ are defined as:*

$$\mathcal{H}_f := \left\{ g \mid g(x) = h_\omega(x)^\top \theta, \ \theta \in \mathbb{R}^p \right\}, \quad \mathcal{H}_{\mathrm{RFF}} := \left\{ g \mid g(x) = \psi_D(x)^\top \theta, \ \theta \in \mathbb{R}^{2D} \right\}, \quad (8)$$

*where $\psi_D = \gamma_D \circ h_\omega$ denotes the vector of composite RFF features defined in Equation 4.*

We will show that $\overline{\mathcal{H}_{\mathrm{RFF}}}$ strictly contains $\mathcal{H}_f$, and thus defines a more expressive hypothesis class. The argument constructs a bridge between the two spaces using a reproducing kernel Hilbert space.

**Theorem 2.** *Let $f$ be any target function in $\mathcal{L}^2(\Omega, \mathbb{R})$. The projection error (see Definition 3 in D.1) achievable by the composite RFF Function Space $\mathcal{H}_{\mathrm{RFF}}$ is no greater than the projection error achieved by the original Feature Space $\mathcal{H}_f$ when the number of RFF features $D$ goes to infinity.*

The proof is given in Appendix D.2. This result establishes a powerful theoretical assurance that RFF embedding offers better approximation capabilities. Theorem 2 yields the universal approximation corollary presented below, the proof of which is given in Appendix D.2.1

**Corollary 1** (Universal approximation). *The projection error of the solution $u$ to equation 1 onto $\mathcal{H}_{\mathrm{RFF}}$ can be made as small as desired, provided enough neurons and RFF features $D$.*

# 5 RELATED WORK

**Weight-balancing strategies** These methods adapt the physics weight $\lambda$ in equation 3 during training. For instance, (Wang et al., 2021a) dynamically updates $\lambda$ to balance the gradients of data and physics losses, while the NTK framework (Jacot et al., 2018; Krishnapriyan et al., 2021) enforces equal decay rates, theoretically recovering high-frequency solutions. Primal–dual methods (Goemans & Williamson, 1997; Barreau & Shen, 2025) instead compute $\lambda$ from the PDE residual. Although simple to implement, these approaches offer weak convergence guarantees and remain tied to single-level optimization. Nonetheless, they are complementary to our framework and could be integrated as weight-balancing strategies within the upper-level problem.

**Resampling strategies** A second line of work reduces the gap between the true loss $\mathfrak{L}_\lambda$ and its sampled counterpart $\hat{\mathfrak{L}}_\lambda$. Examples include NTK-informed sampling (Lau et al., 2024), adversarial sampling (Tang et al., 2024), and reinforcement learning (Song, 2025). While effective in reducing approximation error, these methods do not explicitly target spectral bias, which is the focus of our proposed method.

**Curriculum learning strategies** Finally, new architectures and training schedules aim to better capture high-frequency components. Attention mechanisms (Sirignano & Spiliopoulos, 2018), multi-stage networks (Howard et al., 2025; Waheed, 2022; Chai et al., 2024; Mustajab et al., 2024; Eshkofti & Barreau, 2025; Wang & Lai, 2024), or finite-basis approximation (Moseley et al., 2023) have shown improved multi-scale resolution. However, their complexity often makes training slow and delicate, and they still lack dedicated optimization algorithms.

# 6 NUMERICAL EXPERIMENTS

**Objective.** In this section, we describe comprehensive experiments that establish four main advantages of IFeF-PINN. First, improved approximation over PINNs and SOTA variants on low-frequency PDEs. Second, higher accuracy on high-frequency and multi-scale linear PDEs, where standard PINNs typically show failure modes. Third, our framework exhibits strong generalization capabilities when integrated with advanced PINN variants. Finally, a spectrum analysis experiment demonstrates that our proposed method improves the network fitting accuracy for high-frequency signals.

**Experiment setup.** We will use four PDEs, namely the 2D Helmholtz equation (low and high frequency), 1D convection equation (low and high frequency), 1D convection-diffusion equation, and the viscous Burgers' equation. The baseline methods are Vanilla PINNs, NTK (Wang et al., 2022), PINNsformer (Zhao et al., 2024), and Physics-Informed Gaussians (PIG) (Kang et al., 2025), keeping their default settings for a fair comparison. Additional experimental comparisons with Multiple Fourier Features (MFF) (Wang et al., 2021b) are provided in Appendix G.1. For simplicity, we set $\lambda = 0.01$ for the Vanilla PINNs in Equation 3. Detailed hyperparameters for our proposed methods are in Appendix E. For low-frequency 2D Helmholtz and low-frequency 1D convection equations, we adopt the uniform sampling strategy settings of Zhao et al. (2024). For the viscous Burgers' equation, we follow the setup of Raissi et al. (2019). For the high-frequency Helmholtz equation, we employ Latin hypercube sampling (McKay et al., 2000) to improve domain coverage. We evaluate two variants of our framework: IFeF (Vanilla training) and IFeF-PD (primal-dual weight-balancing proposed by Barreau & Shen (2025)). PDE definitions, datasets, and network architectures are provided in Appendix F. We measure the relative $L^2$-error after convergence, defined as $\frac{\|u_{\text{pred}} - u_{\text{real}}\|_2}{\|u_{\text{real}}\|_2}$. Each method is run five times with independent random seeds, with the best predictions for each approach. All models are implemented in PyTorch and trained on a single NVIDIA GeForce RTX 4090 GPU. The code for all benchmarks is available at https://github.com/CyberAltrumi/IFeF-PINN. Computational aspects are evaluated in Appendix G.2.

## 6.1 RESULTS ON BENCHMARK PDES

We begin with three popular low-frequency benchmark PDEs: 2D Helmholtz equation, 1D convection equation, and the viscous Burgers' equation. Figure 2 summarizes relative $L^2$-errors across baseline methods; box plots display medians and IQRs, and red diamonds denote means. Additional prediction and absolute error maps are provided in Appendix G.

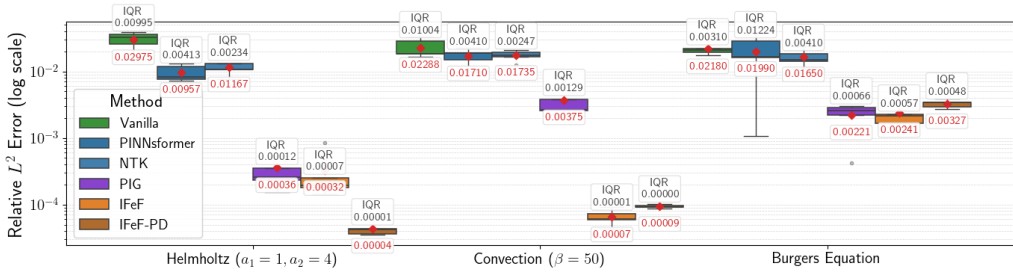

Figure 2: Box plot of relative $L^2$-errors (log10 scale) for all methods on three low-frequency benchmarks with median, inter-quartile range (IQR), and mean (red diamonds).

Across all these problems, our proposed method attains the lowest median errors with reduced variability. On Helmholtz, IFeF-PD achieves the best relative $L^2$ error of $3.5 \times 10^{-5}$. On convection, IFeF achieves the best error of $4.3 \times 10^{-5}$. Even in the nonlinear case of Burgers' equation, IFeF obtains the lowest median error. In addition, we conducted an ablation study where we discarded the RFF basis extension but performed a similar iterative two-step optimization process, obtaining results that were similar but slightly better than those of the Vanilla PINN ($1.4923 \times 10^{-2}$ relative $L^2$-error) on the low-frequency convection problem. Figure 3 presents the predictions for the low-frequency 2D Helmholtz case. On a logarithmic scale, the gap between IFeF-PINN and other methods is consistent with the box plot summaries. These results highlight the strong approximation capability of the proposed method, especially for linear equations, underscoring its robustness for solving diverse PDEs.

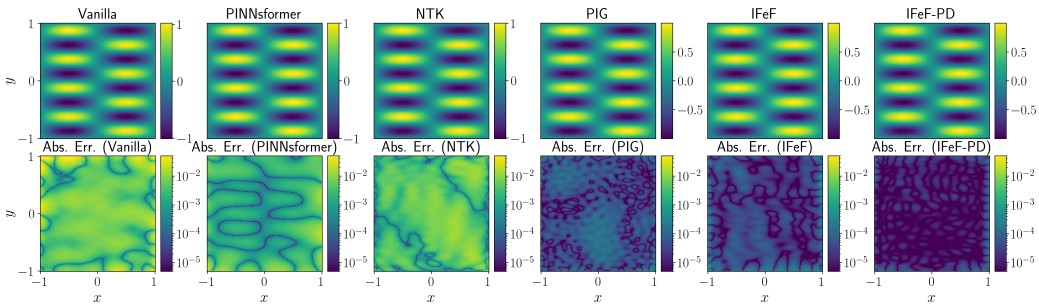

Figure 3: Low-frequency Helmholtz equation prediction solution (up) and absolute error on a log10 scale (bottom) of baseline methods.

## 6.2 MITIGATING THE SPECTRAL BIAS

To evaluate challenging cases of spectral bias, we study the failure modes of PINNs on high-frequency and multi-scale PDEs, where vanilla PINNs typically struggle to learn rapidly oscillatory or widely separated frequency components. In particular, we study the high-frequency Helmholtz and convection equations, as well as a multi-scale convection-diffusion equation. Table 2 presents the mean and standard deviation of the relative $L^2$-errors over baselines applied to these problems. Additional prediction and absolute error maps are provided in Appendix G.

| Baseline | Helmholtz $(a_1 = a_2 = 100)$ | Convection $(\beta = 200)$ | Convection-Diffusion $(k_{\text{low}} = 4\pi, k_{\text{high}} = 60\pi)$ |
|---|---|---|---|
| Vanilla | - | 0.9024 (0.0239) | 0.0501 (0.0030) |
| PINNsformer | - | 1.2278 (0.2010) | 0.0525 (0.0001) |
| NTK | - | 0.8685 (0.0318) | 0.0526 (0.0001) |
| PIG | 1.6884 (0.2775) | 1.0009 (0.0003) | 0.0560 (0.0010) |
| IFeF | 0.0156 (0.0055) | 0.0027 (0.0010) | **0.0009** (0.0003) |
| IFeF-PD | **0.0092** (0.0031) | **0.0025** (0.0005) | 0.0010 (0.0002) |

Table 2: Average relative $L^2$-error with corresponding standard deviation for each baseline on three high-frequency PDEs. A dash '-' denotes that the baseline failed to converge.

Figure 4 depicts the high-frequency Helmholtz solutions and the corresponding log-scale absolute errors. In the considered scenarios, all baselines exhibit clear failure modes. We also conducted a similar ablation study as described in the previous section, removing the RFF basis extension, and the training did not converge for both the high-frequency Helmholtz and convection equations. In contrast, the proposed IFeF-PINN method effectively mitigates the spectral bias of neural networks. Moreover, when combined with the primal-dual method to adaptively balance the physics-based loss, our method achieves accurate solutions even under very high frequencies, which illustrates the flexibility of the proposed framework in incorporating advanced learning methods. A similar result holds for the multi-scale convection-diffusion equation in Figure 4 in Appendix G, clearly showing that only IFeF-PINN succeeds in learning both low and high frequency components of the solution.

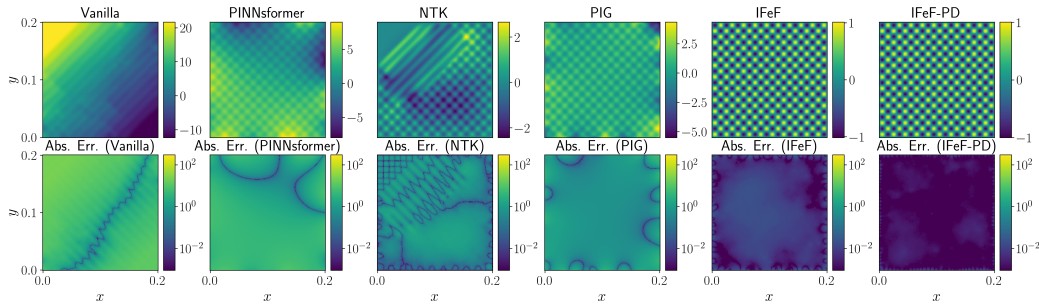

Figure 4: High-frequency Helmholtz equation prediction solution (up) and absolute error in log scale (bottom) of baseline methods.

In contrast, all baselines suffer from the spectral bias failure mode, where models prioritize learning low-frequency components and tend to ignore the high-frequency components.

## 6.3 SPECTRUM ANALYSIS

To quantitatively demonstrate our method's ability to mitigate spectral bias, we employ the fast Fourier transform to analyze the frequency-domain distribution of the network's prediction. We conduct a spectrum analysis similar to Rahaman et al. (2019), designing a challenging multi-scale convection equation with an initial condition composed of a superposition of ten sinusoids of different frequencies and unit amplitude. More details of the setup are in Appendix F.2.

During analysis, we compare the performance of Vanilla PINNs against models where the basis is extended with a varying number of random Fourier features and carry out a one-step solution of the lower-level objective in Equation 6. No additional training is performed for the upper-level problem.

We compute the magnitude of their discrete Fourier transform at frequencies $k_i$, denoted as $|\tilde{f}_{k_i}|$. Figure 5 presents the average normalized magnitudes $\frac{|\tilde{f}_{k_i}|}{A_i}$ over five independent runs. The results clearly illustrate the spectral bias of Vanilla PINNs, which struggle to accurately capture high-frequency components. In contrast, by extending the network's basis through RFF, the network can fit high-frequency signals much more effectively, even without the sub-

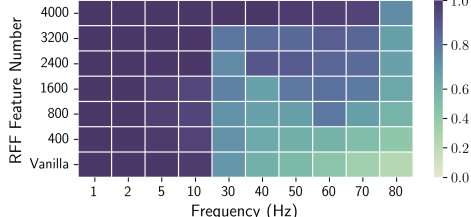

Figure 5: Prediction of the network spectrum with an increasing number of Fourier features. The x-axis represents frequency, and the colorbar shows the normalized magnitude of the predicted solution at $t = 0$. The colorbar is scaled accordingly from 0 to 1.

sequent bi-level training procedure of IFeF-PINN. Furthermore, we observe that increasing the number of random features enhances the network's ability to approximate high-frequency components, confirming the effectiveness of our basis extension strategy.

## 6.4 ABLATION STUDIES

In this section, we present experiments to demonstrate the effects of two-stage training in IFeF-PINN and the number of Fourier-enhanced features and the Gaussian sampling parameter $\sigma$.

### 6.4.1 END-TO-END TRAINING

To validate the necessity of two-stage training in IFeF-PINN, we conduct an end-to-end ablation where both network parameters $\omega$ and coefficients $\theta$ are jointly optimized. We keep the approximation in Equation 5 but incorporate $\theta$ as learnable parameters alongside $\omega$, and directly minimize $\hat{\mathfrak{L}}_\lambda(u_{\omega,\theta})$ without the two-stage training. Unlike IFeF-PINN where $\theta$ is always optimal under current features $\psi_D(x)$, $\theta$ is randomly initialized and updated simultaneously with $\omega$, losing the optimality guarantee. Table 3 presents the results for low- and high-frequency Helmholtz and Burgers' equa-

tions. This ablation validates the necessity of two-stage training, as IFeF-PINN significantly outperforms end-to-end training in linear PDEs through guaranteed lower-level optimality of $\theta$, while showing modest improvements in nonlinear PDEs where the lower-level becomes non-convex.

| Ablation | Helmholtz $(a_1 = 1, a_2 = 4)$ | Helmholtz $(a_1 = a_2 = 100)$ | Viscous Burgers $(\nu = \frac{0.01}{\pi})$ |
|---|---|---|---|
| End-to-End | 0.0088(0.0006) | - | 0.0049(0.0009) |
| IFeF | 0.0003(0.0003) | 0.0156(0.0055) | 0.0024(0.0011) |
| IFeF-PD | 0.00005(0.00002) | 0.0092(0.0031) | 0.0033(0.0004) |

Table 3: Average relative $L^2$-error with corresponding standard deviation for end-to-end training and IFeF-PINN on three benchmarks. A dash '-' denotes that the baseline failed to converge.

### 6.4.2 HYPERPARAMETER ABLATION

We conduct an ablation on two key hyperparameters in IFeF-PINN: the number of Fourier features $D$ and the Gaussian sampling parameter $\sigma$. We evaluate their impact on performance using the low- and high-frequency Helmholtz equations, with results shown in Table 4. The ablation shows that too few features reduce expressivity while excessive features cause overfitting and may break the rank condition discussed in Appendix B.1. For $\sigma$, larger values are essential for high-frequency problems discussed in Tancik et al. (2020); Wang et al. (2021b). Low-frequency problems are robust to both hyperparameters, while high-frequency problems are sensitive, especially to $\sigma$.

| Helmholtz $(a_1 = 1, a_2 = 4)$ | | | | | |
|---|---|---|---|---|---|
| $D$ ($\sigma = 1$) | 100 | 400 | 800 | 1200 | 3000 |
| Rel. $L^2$ error | $5.5 \times 10^{-4}$ | $\mathbf{2.1 \times 10^{-4}}$ | $3.2 \times 10^{-4}$ | $5.7 \times 10^{-4}$ | $4.5 \times 10^{-4}$ |
| $\sigma$ ($D = 800$) | 2 | 1 | 0.5 | 0.2 | 0.1 |
| Rel. $L^2$ error | $4.0 \times 10^{-4}$ | $\mathbf{3.2 \times 10^{-4}}$ | $5.5 \times 10^{-4}$ | $3.3 \times 10^{-4}$ | $1.5 \times 10^{-3}$ |
| Helmholtz $(a_1 = a_2 = 100)$ | | | | | |
| $D$ ($\sigma = 1$) | 800 | 1200 | 1600 | 2400 | 3000 |
| Rel. $L^2$ error | $7.11 \times 10^{-2}$ | $5.40 \times 10^{-2}$ | $3.09 \times 10^{-2}$ | $\mathbf{1.56 \times 10^{-2}}$ | $2.22 \times 10^{-2}$ |
| $\sigma$ ($D = 2400$) | 20 | 10 | 5 | 1 | 0.2 |
| Rel. $L^2$ error | $4.6 \times 10^{-3}$ | $\mathbf{3.0 \times 10^{-3}}$ | $5.7 \times 10^{-3}$ | $1.56 \times 10^{-2}$ | $1.05 \times 10^{-1}$ |

Table 4: Average relative $L^2$-error for hyperparameter ablation for $D$ and $\sigma$ on Helmholtz equations.

## 7 CONCLUSION

In this paper, we introduce IFeF-PINN, a novel iterative training method for Fourier-enhanced Features PINNs. By augmenting the network with random Fourier features mapping as a basis extension with the bi-level problem, IFeF-PINN mitigates the spectral bias problem of standard PINNs when capturing the high-frequency and multi-scale components during training. Experimental results demonstrate that IFeF-PINN consistently outperforms advanced baselines across various scenarios, including popular low-frequency benchmarks and handling high-frequency and multi-scale PDEs. Furthermore, it has strong flexibility when integrating with different training strategies for PINNs.

Despite its strengths, IFeF-PINN faces challenges when extended to nonlinear PDEs. For nonlinear PDEs, the lower-level problem becomes nonconvex, precluding a one-step solve and requiring iterative two-stage gradient descent updates that can stall in local minima. Advancing principled bi-level optimization techniques to better handle the nonlinear lower-level problem remains a promising direction for future work.

## 8 ACKNOWLEDGMENTS

This work was partially supported by the Wallenberg AI, Autonomous Systems and Software Program (WASP), funded by the Knut and Alice Wallenberg Foundation. It was further supported by the Swedish Research Council through the Distinguished Professor Grant 2017-01078, as well as by the Wallenberg Scholar Grant from the Knut and Alice Wallenberg Foundation. The authors also gratefully acknowledge the support of Digital Futures.

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
