# Iterative Training of Physics-Informed Neural Networks with Fourier-enhanced Features

**Yulun Wu, Miguel Aguiar, Karl H. Johansson & Matthieu Barreau**
Division of Decision and Control Systems
Digital Futures and KTH Royal Institute of Technology
Stockholm, Sweden
{yulunw,aguiar,kallej,barreau}@kth.se

## A  Contents

We organize this supplementary document as follows:

## B  QP formulation details and proof of Proposition 1

Let $X_u = (x_u^1, \ldots x_u^{N_u}) \in \mathbb{R}^{N_u \times n}$ be boundary collocation points and $X_f = (x_f^1, \ldots, x_f^{N_f}) \in \mathbb{R}^{N_f \times n}$ be interior collocation points. We define

- $B_u(\omega) \in \mathbb{R}^{N_u \times 2D}$ with rows $\mathfrak{B}[\psi \circ h_\omega](x_u^i)^\top \in \mathbb{R}^n$ for $i = 1, \ldots, N_u$ as boundary values;
- $G_u \in \mathbb{R}^{N_u}$ with rows $g(x_u^i)^\top \in \mathbb{R}^n$ for $i = 1, \ldots, N_u$ as boundary measurement;
- $R_f(\omega) \in \mathbb{R}^{N_f \times 2D}$ with rows $\mathfrak{F}[\psi \circ h_\omega](x_f^i)^\top$ for $i = 1, \ldots, N_f$ as residual values;
- $F_f \in \mathbb{R}^{N_f}$ with rows $f(x_f^i)^\top$ for $i = 1, \ldots, N_f$ as source terms.

Note that the linearity of the $\mathfrak{B}$ and $\mathfrak{F}$ operators leads to $\mathfrak{B}[u_{\omega,\theta}](X_u) = B_u \theta$ and $\mathfrak{F}[u_{\omega,\theta}](X_f) = R_f \theta$. Consequently, using the standard PIML loss with boundary and PDE residual terms, we get:

$$\hat{\mathfrak{L}}_\lambda(u_{\omega,\theta}) = \frac{1}{N_u} \big\| B_u(\omega)\theta - G_u \big\|^2 + \frac{\lambda}{N_f} \big\| R_f(\omega)\theta - F_f \big\|^2. \tag{1}$$

The loss expands to the quadratic form

$$\mathfrak{L}_{\text{lower}}(\theta \mid \omega) = \frac{1}{2}\theta^\top \left( \frac{2}{N_u} B_u^\top B_u + \frac{2\lambda_{\text{LL}}}{N_f} R_f^\top R_f \right)\theta + \left( -\frac{2}{N_u} B_u^\top G_u - \frac{2\lambda_{\text{LL}}}{N_f} R_f^\top F_f \right)^\top \theta$$
$$+ \frac{1}{N_u} G_u^\top G_u + \frac{\lambda_{\text{LL}}}{N_f} F_f^\top F_f. \tag{2}$$

Identifying

$$Q(\omega) = \frac{2}{N_u} B_u(\omega)^\top B_u(\omega) + \frac{2\lambda_{\text{LL}}}{N_f} R_f(\omega)^\top R_f(\omega), \quad c(\omega) = -\frac{2}{N_u} B_u(\omega)^\top G_u - \frac{2\lambda_{\text{LL}}}{N_f} R_f(\omega)^\top F_f,$$

leads to $\mathfrak{L}_{\text{lower}}(\theta \mid \omega) = \frac{1}{2}\theta^\top Q(\omega)\theta + c(\omega)^\top \theta + b$, where $b$ is a constant term and $\lambda_{\text{LL}}$ is the physics weight used in the lower-level problem.

## B.1 Analysis of the Rank Condition and Regularization of Proposition 1

The positive semi-definiteness of $Q(\omega)$ follows from the factorization $Q(\omega) = M(\omega)^\top M(\omega) \in \mathbb{R}^{2D \times 2D}$, with the stacked design matrices $M(\omega)$ as follows:

$$M(\omega) = \begin{pmatrix} \sqrt{\frac{2}{N_u}} B_u(\omega) \\ \sqrt{\frac{2\lambda}{N_f}} R_f(\omega) \end{pmatrix},$$

where $\lambda > 0$ is always ensured. Then we have $\mathrm{rank}(Q(\omega)) = \mathrm{rank}(M(\omega))$. Hence, $Q(\omega)$ is strictly positive definite if and only if $M(\omega)$ has full column rank, i.e.,

$$\mathrm{rank}(M(\omega)) = 2D. \tag{3}$$

The loss $\mathcal{L}_{\mathrm{lower}}(\theta \mid \omega)$ becomes then strongly convex, and its minimization has a unique solution $\theta^\star = -Q^{-1}c$.

The rank condition in equation 3 imposes a fundamental constraint: for the stacked design matrix $M \in \mathbb{R}^{(N_u + N_f) \times 2D}$ to have full column rank $2D$, it is necessary that the number of rows is at least as large as the number of columns. This leads to the critical requirement on the number of sampling points: $N_u + N_f \geq 2D$.

When this condition is violated (i.e., $N_u + N_f < 2D$), the system becomes underdetermined. This directly causes the matrix $Q(\omega)$ to be rank-deficient. Consequently, the lower-level problem becomes ill-posed, lacking a unique solution as $Q^{-1}$ does not exist. This might lead to an aliased solution (especially true when we add a Tikhonov regularization).

A more subtle cause of rank deficiency occurs even when $N_u + N_f \geq 2D$. This happens if the collocation points provide redundant information, failing to create $2D$ linearly independent constraints. Such a situation can arise from geometrically poor sampling (e.g., points lying on nodal lines of the basis functions) or from inherent redundancies in the randomly generated RFF basis itself (e.g., two different random vectors being nearly parallel). In these scenarios, although the design matrix $M$ has enough rows, its columns remain linearly dependent, leading to a singular or, more commonly, a numerically ill-conditioned matrix $Q$. However, if $D$ is large, a solution might be to just discard these redundancies while still keeping the same feature space $\mathcal{H}_{RFF}$ (see equation 8 of the main paper).

To resolve this, we employ Tikhonov regularization, modifying the matrix to $Q_{\mathrm{reg}}(\omega) = Q(\omega) + \gamma I$, where $\gamma > 0$ is a small regularization parameter. This ensures $Q_{\mathrm{reg}}$ is always positive definite and invertible, since for any non-zero $\theta$, the quadratic form $\theta^\top Q_{\mathrm{reg}}\theta = \theta^\top Q\theta + \gamma\|\theta\|^2$ is strictly positive. The lower-level problem thus regains a unique, stable solution

$$\theta^\star(\omega) = -(Q(\omega) + \gamma I)^{-1}c(\omega). \tag{4}$$

This analysis reveals a fundamental trade-off: while increasing $D$ enhances representational power, it demands proportionally more sampling points to maintain a well-posed system. When the sampling budget is limited, Tikhonov regularization provides a principled remedy to ensure algorithmic stability, at the cost of introducing a slight bias to the solution.

## C Proofs of Convergence Analysis

### C.1 Hypergradient Derivation via Implicit Function Theorem (IFT)

The hypergradient $\nabla_\omega \mathcal{L}_{\mathrm{upper}}(\omega)$ is computed using the chain rule, where the Implicit Function Theorem (IFT) provides the Jacobian of the lower-level solution map $\theta^\star(\omega)$.

The lower-level optimality condition is $F(\theta^\star, \omega) := Q(\omega)\theta^\star + c(\omega) = 0$. Taking the total derivative with respect to $\omega$ yields

$$\frac{\partial F}{\partial \theta^\top} \frac{\partial \theta^\star}{\partial \omega^\top} + \frac{\partial F}{\partial \omega^\top} = 0.$$

Solving for the Jacobian $\frac{\partial \theta^\star}{\partial \omega^\top}$ gives

$$\frac{\partial \theta^\star}{\partial \omega^\top} = -\left(\frac{\partial F}{\partial \theta^\top}\right)^{-1} \frac{\partial F}{\partial \omega^\top} = -Q(\omega)^{-1}\left(\frac{\partial(Q(\omega)\theta^\star)}{\partial \omega^\top} + \frac{\partial c(\omega)}{\partial \omega^\top}\right).$$

The full hypergradient is then obtained by the chain rule

$$\nabla_\omega \mathfrak{L}_{\text{upper}}(\omega) = \frac{\partial \mathfrak{L}_{\text{upper}}}{\partial \omega} + \left( \frac{\partial \theta^\star}{\partial \omega^\top} \right)^\top \frac{\partial \mathfrak{L}_{\text{upper}}}{\partial \theta}. \tag{5}$$

Substituting the expression for the Jacobian, we get

$$\nabla_\omega \mathfrak{L}_{\text{upper}}(\omega) = \frac{\partial \mathfrak{L}_{\text{upper}}}{\partial \omega} - \left( \frac{\partial (Q(\omega)\theta^*)}{\partial \omega^\top} + \frac{\partial c(\omega)}{\partial \omega^\top} \right)^\top Q(\omega)^{-1} \frac{\partial \mathfrak{L}_{\text{upper}}}{\partial \theta}.$$

This provides a computable formula for the gradient used in the upper-level optimization.

## C.2 PROOF OF PROPOSITION 2

We first state the key properties required for our convergence analysis.

*Assumption* 1 (Smoothness and Boundedness Properties). The bi-level optimization problem satisfies the following regularity conditions:

1. The functions $Q(\omega)$ and $c(\omega)$ are continuously differentiable with respect to $\omega$. The upper-level loss $\mathfrak{L}_{\text{upper}}(\theta, \omega)$ is continuously differentiable with respect to both $\theta$ and $\omega$.

2. The lower-level problem is $\mu$-strongly convex, i.e., $Q(\omega) \succeq \mu I$ for some constant $\mu > 0$.

3. The objective function $\mathfrak{L}_{\text{upper}}(\omega)$ is bounded below by a scalar $\mathfrak{L}_{\text{inf}}$.

*Assumption* 2 (L-Smoothness of the Hypergradient). The upper-level objective function $\mathfrak{L}_{\text{upper}}(\omega)$ is L-smooth, a standard assumption in gradient-based optimization analysis. This means its gradient, the hypergradient $\nabla_\omega \mathfrak{L}_{\text{upper}}(\omega)$, is Lipschitz continuous with constant $L > 0$:

$$\|\nabla_\omega \mathfrak{L}_{\text{upper}}(\omega_1) - \nabla_\omega \mathfrak{L}_{\text{upper}}(\omega_2)\| \leq L\|\omega_1 - \omega_2\|, \quad \forall \omega_1, \omega_2 \in \mathbb{R}^P. \tag{6}$$

These assumptions trivially hold when neural network activation functions are Lipschitz continuous and the loss function is smooth, which is satisfied by our choice of $\tanh$ activations with MSE losses.

*Proof.* Since $\mathfrak{L}_{\text{lower}}$ is strongly convex and differentiable with respect to $\theta$, the unique optimal solution $\theta^\star(\omega)$ is found by setting the gradient to zero:

$$\nabla_\theta \mathfrak{L}_{\text{lower}}(\theta^\star(\omega)) = Q(\omega)\theta^\star(\omega) + c(\omega) = 0. \tag{7}$$

This gives the closed-form solution showed in Proposition 1:

$$\theta^\star(\omega) = -Q(\omega)^{-1}c(\omega). \tag{8}$$

Then consider two parameter vectors $\omega_1, \omega_2$ from a compact set $\mathcal{W}$. We want to bound the norm of the difference $\|\theta^\star(\omega_1) - \theta^\star(\omega_2)\|$. This follows the structure from your provided image:

$$\begin{aligned}
\|\theta^\star(\omega_1) - \theta^\star(\omega_2)\| &= \|Q(\omega_1)^{-1}c(\omega_1) - Q(\omega_2)^{-1}c(\omega_2)\| \\
&= \|Q(\omega_1)^{-1}c(\omega_1) - Q(\omega_1)^{-1}c(\omega_2) + Q(\omega_1)^{-1}c(\omega_2) - Q(\omega_2)^{-1}c(\omega_2)\| \\
&\leq \|Q(\omega_1)^{-1}(c(\omega_1) - c(\omega_2))\| + \|(Q(\omega_1)^{-1} - Q(\omega_2)^{-1})c(\omega_2)\| \\
&\leq \|Q(\omega_1)^{-1}\| \cdot \|c(\omega_1) - c(\omega_2)\| + \|Q(\omega_1)^{-1} - Q(\omega_2)^{-1}\| \cdot \|c(\omega_2)\|. \quad (9)
\end{aligned}$$

We use the matrix identity $A^{-1} - B^{-1} = A^{-1}(B - A)B^{-1}$ to bound the second term:

$$\begin{aligned}
\|Q(\omega_1)^{-1} - Q(\omega_2)^{-1}\| &= \|Q(\omega_1)^{-1}(Q(\omega_2) - Q(\omega_1))Q(\omega_2)^{-1}\| \\
&\leq \|Q(\omega_1)^{-1}\| \cdot \|Q(\omega_2) - Q(\omega_1)\| \cdot \|Q(\omega_2)^{-1}\|. \quad (10)
\end{aligned}$$

Substituting Equation 10 back into Equation 9:

$$\begin{aligned}
\|\theta^\star(\omega_1) - \theta^\star(\omega_2)\| &\leq \|Q(\omega_1)^{-1}\| \cdot \|c(\omega_1) - c(\omega_2)\| \\
&\quad + \|Q(\omega_1)^{-1}\| \cdot \|Q(\omega_2) - Q(\omega_1)\| \cdot \|Q(\omega_2)^{-1}\| \cdot \|c(\omega_2)\|. \quad (11)
\end{aligned}$$

By Assumption 1, on the compact set $\mathcal{W}$, there exist constants $L_Q, L_c > 0$ such that $\|Q(\omega_1) - Q(\omega_2)\| \leq L_Q \|\omega_1 - \omega_2\|$ and $\|c(\omega_1) - c(\omega_2)\| \leq L_c \|\omega_1 - \omega_2\|$. Furthermore, due

to strong convexity, there is a $\mu_Q > 0$ such that $\left\|Q(\omega)^{-1}\right\| \leq 1/\mu_Q$ for all $\omega \in \mathcal{W}$. Finally, since $c(\omega)$ is continuous on a compact set, its norm is bounded by a constant $C_{\max} = \sup_{\omega \in \mathcal{W}} \|c(\omega)\|$.

Substituting these bounds into Equation 11:

$$\|\theta^\star(\omega_1) - \theta^\star(\omega_2)\| \leq \frac{1}{\mu_Q}(L_c \|\omega_1 - \omega_2\|) + \left(\frac{1}{\mu_Q} \cdot L_Q \|\omega_1 - \omega_2\| \cdot \frac{1}{\mu_Q}\right) C_{\max}$$

$$= \left(\frac{L_c}{\mu_Q} + \frac{L_Q C_{\max}}{\mu_Q^2}\right) \|\omega_1 - \omega_2\|.$$

Defining the constant $K = \frac{L_c}{\mu_Q} + \frac{L_Q C_{\max}}{\mu_Q^2}$ completes the proof. $\qquad\square$

### C.3 PROOF OF THEOREM 1

The proof relies on the following standard lemma for L-smooth functions.

**Lemma 1** (Sufficient Decrease). *If $\mathfrak{L}_{\text{upper}}(\omega)$ is L-smooth with constant $L$ and the step size $\eta \in (0, 2/L)$, the gradient descent update rule ensures a sufficient decrease in the objective function:*

$$\mathfrak{L}_{\text{upper}}(\omega_{k+1}) \leq \mathfrak{L}_{\text{upper}}(\omega_k) - \eta \left(1 - \frac{L\eta}{2}\right) \|\nabla_\omega \mathfrak{L}_{\text{upper}}(\omega_k)\|^2.$$

*Proof.* From the L-smoothness property (descent lemma) under Assumption 2, we have:

$$\mathfrak{L}_{\text{upper}}(\omega_{k+1}) \leq \mathfrak{L}_{\text{upper}}(\omega_k) + \nabla_\omega \mathfrak{L}_{\text{upper}}(\omega_k)^\top (\omega_{k+1} - \omega_k) + \frac{L}{2} \|\omega_{k+1} - \omega_k\|^2$$

Substituting the gradient descent update $\omega_{k+1} - \omega_k = -\eta \nabla_\omega \mathfrak{L}_{\text{upper}}(\omega_k)$:

$$\mathfrak{L}_{\text{upper}}(\omega_{k+1}) \leq \mathfrak{L}_{\text{upper}}(\omega_k) - \eta \|\nabla_\omega \mathfrak{L}_{\text{upper}}(\omega_k)\|^2 + \frac{L\eta^2}{2} \|\nabla_\omega \mathfrak{L}_{\text{upper}}(\omega_k)\|^2$$

$$= \mathfrak{L}_{\text{upper}}(\omega_k) - \eta \left(1 - \frac{L\eta}{2}\right) \|\nabla_\omega \mathfrak{L}_{\text{upper}}(\omega_k)\|^2.$$

$\qquad\square$

*Proof of Theorem 1.* Let $\delta = \eta(1 - L\eta/2)$. Since $\eta \in (0, 2/L)$, we have $\delta > 0$. Rearranging the inequality from Lemma 1 gives:

$$\delta \|\nabla_\omega \mathfrak{L}_{\text{upper}}(\omega_k)\|^2 \leq \mathfrak{L}_{\text{upper}}(\omega_k) - \mathfrak{L}_{\text{upper}}(\omega_{k+1}).$$

We now sum this inequality from $k = 0$ to $T$ to form a telescoping series:

$$\sum_{k=0}^{T} \delta \|\nabla_\omega \mathfrak{L}_{\text{upper}}(\omega_k)\|^2 \leq \sum_{k=0}^{T} (\mathfrak{L}_{\text{upper}}(\omega_k) - \mathfrak{L}_{\text{upper}}(\omega_{k+1}))$$

$$= (\mathfrak{L}_{\text{upper}}(\omega_0) - \mathfrak{L}_{\text{upper}}(\omega_1)) + (\mathfrak{L}_{\text{upper}}(\omega_1) - \mathfrak{L}_{\text{upper}}(\omega_2)) + \ldots$$

$$+ (\mathfrak{L}_{\text{upper}}(\omega_T) - \mathfrak{L}_{\text{upper}}(\omega_{T+1}))$$

$$= \mathfrak{L}_{\text{upper}}(\omega_0) - \mathfrak{L}_{\text{upper}}(\omega_{T+1}).$$

The objective function is bounded below by $\mathfrak{L}_{\inf} \geq 0$. Therefore, $\mathfrak{L}_{\text{upper}}(\omega_{T+1}) \geq \mathfrak{L}_{\inf}$. This gives us:

$$\sum_{k=0}^{T} \delta \|\nabla_\omega \mathfrak{L}_{\text{upper}}(\omega_k)\|^2 \leq \mathfrak{L}_{\text{upper}}(\omega_0) - \mathfrak{L}_{\inf}.$$

As $T \to \infty$, the right-hand side is a finite constant. This implies that the infinite series of squared gradient norms is bounded:

$$\sum_{k=0}^{\infty} \|\nabla_\omega \mathfrak{L}_{\text{upper}}(\omega_k)\|^2 \leq \frac{\mathfrak{L}_{\text{upper}}(\omega_0) - \mathfrak{L}_{\inf}}{\delta} < \infty.$$

For an infinite series of non-negative terms to converge to a finite value, the terms themselves must converge to zero. Therefore, we must have:

$$\lim_{k \to \infty} \|\nabla_\omega \mathfrak{L}_{\text{upper}}(\omega_k)\|^2 = 0,$$

which implies that $\lim_{k \to \infty} \|\nabla_\omega \mathfrak{L}_{\text{upper}}(\omega_k)\| = 0$. This completes the proof that the algorithm converges to a stationary point. $\square$

## D  PROOFS FOR PROJECTION ERROR ANALYSIS

This appendix provides the foundational definitions and detailed proofs for the projection error analysis presented in Section 4.2.

### D.1  FOUNDATIONAL CONCEPTS

**Definition 1** (Universal Kernel). *A continuous kernel $k$ defined on a compact metric space $(\mathcal{X}, d)$ is called a **universal kernel** if the Reproducing Kernel Hilbert Space (RKHS) $\mathcal{H}_k$ induced by $k$ is dense in the space of continuous functions $C(\mathcal{X})$ with respect to the uniform norm $\|\cdot\|_\infty$.*

*Mathematically, this means that for any function $g \in C(\mathcal{X})$ and any $\varepsilon > 0$, there exists a function $f \in \mathcal{H}_k$ such that:*

$$\sup_{x \in \mathcal{X}} |f(x) - g(x)| < \varepsilon.$$

*An equivalent way to state this is that the closure of $\mathcal{H}_k$ under the uniform norm is $C(\mathcal{X})$:*

$$\overline{\mathcal{H}_k} = C(\mathcal{X}).$$

**Definition 2.** *Let $f \in \mathcal{L}^2(\Omega, \mathbb{R})$ be a target function. The **projection error** of $f$ onto an Hilbert space $\mathcal{H} \subseteq \mathcal{L}^2(\Omega, \mathbb{R})$ is defined as*

$$\text{Err}(f, \mathcal{H}) := \inf_{g \in \mathcal{H}} \|f - g\|. \tag{12}$$

*If this infimum is attained by some $g^* \in \mathcal{H}$, then $g^*$ is the projection of $f$ onto $\mathcal{H}$.*

**Theorem 1** (Composition of Universal Kernels). *If $k(z, z')$ is a universal kernel on a space $\mathcal{Z}$, and the mapping $h : \mathcal{X} \to \mathcal{Z}$ is continuous and sufficiently expressive (e.g., injective), then the composite kernel $k_h(x, x') := k(h(x), h(x'))$ is universal on $\mathcal{X}$.*

*Remark* 1. The universality of the composite kernel relies on the composition theorem from Micchelli et al. (2006).

**Theorem 2** (RFF Approximation). *Let $k : \mathbb{R}^m \times \mathbb{R}^m \to \mathbb{R}$ be a continuous, translation-invariant, positive definite kernel function, i.e.,*

$$k(x, x') = k(x - x') = \int_{\mathbb{R}^d} e^{iw^T(x - x')} d\mu(w),$$

*where $\mu$ is a probability measure with compact support. Define the Random Fourier Feature (RFF) map $\phi_{\text{RFF}} : \mathcal{X} \to \mathbb{R}^{2D}$ as*

$$\phi_{\text{RFF}}(x) := \sqrt{\frac{1}{D}} \begin{bmatrix} \cos(w_1^\top x + b_1) \\ \cos(w_D^\top x + b_D) \\ \vdots \\ \sin(w_1^\top x + b_1) \\ \sin(w_D^\top x + b_D) \end{bmatrix}, \tag{13}$$

*where $w_i \overset{\text{i.i.d.}}{\sim} \mu$ and $b_i \overset{\text{i.i.d.}}{\sim} \text{Uniform}[0, 2\pi]$, with $\{w_i\}$ independent of $\{b_i\}$.*

*Let $\mathcal{H}_k$ be the RKHS corresponding to $k$. For any $f \in \mathcal{H}_k$ and any probability distribution $\rho$:*

$$\lim_{D \to \infty} \inf_{\theta \in \mathbb{R}^D} \|f - \phi_{\text{RFF}}^T \theta\|_{L^2(\rho)} = 0,$$

*i.e., the function space spanned by RFF features is dense in $\mathcal{H}_k$.*

*Remark* 2. This result is a direct consequence of the uniform convergence of the RFF kernel approximation to the true kernel Rahimi & Recht (2007).

**Theorem 3** (RFF Approximation for Composite Kernels). *Let $k$ be a continuous, translation-invariant, positive definite kernel function on $\mathbb{R}^m$ and $h : \mathbb{R}^d \to \mathbb{R}^m$ be a continuous mapping. Let $\mathcal{H}_{k_h}$ be the RKHS of the composite kernel $k_h(x, x') = k(h(x), h(x'))$. The function space spanned by the composite RFF features, $\mathcal{H}_{\mathrm{RFF}}$ defined in Equation 8 of the main paper, is dense in $\mathcal{H}_{k_h}$ with respect to the $L^2(\rho)$ norm when $D \to \infty$.*

$$\lim_{D \to \infty} \inf_{\theta \in \mathbb{R}^D} \left\| f(x) - \psi(x)^\top \theta \right\|_{L^2(\rho)} = 0$$

*Proof.* The proof connects the approximation properties in the base space $\mathcal{H}_k$ to the composite space $\mathcal{H}_{k_h}$ through the mapping $h$. The composite RKHS $\mathcal{H}_{k_h}$ consists of functions formed by composing elements from the base RKHS $\mathcal{H}_k$ with the mapping $h$, that is, $\mathcal{H}_{k_h} = \{g(h(\cdot)) \mid g \in \mathcal{H}_k\}$. Thus, for any function $f \in \mathcal{H}_{k_h}$, there exists a corresponding function $g \in \mathcal{H}_k$ such that $f(x) = g(h(x))$ for all $x \in \mathbb{R}^d$.

To proceed, define a standard Random Fourier Feature (RFF) map for the base kernel $k(z, z')$ on $\mathbb{R}^m$:

$$\psi_D(z) := \sqrt{\frac{1}{D}} \begin{bmatrix} \cos(w_1^T z) \\ \vdots \\ \cos(w_D^T z) \\ \sin(w_1^T z) \\ \vdots \\ \sin(w_D^T z) \end{bmatrix},$$

where $w_i \overset{\text{i.i.d.}}{\sim} \mu$ and $b_i \overset{\text{i.i.d.}}{\sim} \mathrm{Uniform}[0, 2\pi]$, with $\{w_i\}$ independent of $\{b_i\}$. The classic RFF approximation (Theorem 2) guarantees that the linear span of these features is dense in $\mathcal{H}_k$ with respect to the $L^2$ norm under suitable measures. Let $\rho_h$ be the pushforward probability measure of $\rho$ under the map $h$. Then, for the function $g \in \mathcal{H}_k$,

$$\lim_{D \to \infty} \inf_{\theta \in \mathbb{R}^{2D}} \left\| g - \psi_D(x)^T \theta \right\|_{L^2(\rho_h)} = 0.$$

The goal is to show that $f(x)$ can be approximated by a function of the form $\psi(x)^\top \theta$. We analyze the squared $L^2(\rho)$ norm of the error:

$$\left\| f(x) - \psi(x)^\top \theta \right\|_{L^2(\rho)}^2 = \int_{\mathbb{R}^d} \left| f(x) - \psi(x)^\top \theta \right|^2 d\rho(x).$$

Substituting $f(x) = g(h(x))$ and $\psi(x) = \psi_D(h(x))$ yields

$$\int_{\mathbb{R}^d} \left| g(h(x)) - \psi_D(h(x))^T \theta \right|^2 d\rho(x).$$

By the change of variables (or pushforward measure property), this integral equals

$$\int_{\mathbb{R}^m} \left| g(z) - \psi_D(z)^T \theta \right|^2 d\rho_h(z) = \left\| g - \psi_D^T \theta \right\|_{L^2(\rho_h)}^2.$$

From the density in $L^2(\rho_h)$, this error can be made arbitrarily small as $D \to \infty$ by choosing appropriate $\theta$. Therefore, for any $f \in \mathcal{H}_{k_h}$,

$$\lim_{D \to \infty} \inf_{\theta \in \mathbb{R}^{2D}} \left\| f - \psi_D^T \theta \right\|_{L^2(\rho)} = 0,$$

completing the proof. $\qquad \square$

**Lemma 2** (Expressive Power of the RFF-Enhanced Space). *When the number of the RFF features $D \to \infty$, the Feature Space $\mathcal{H}_f$ is a subset of the closure of the Composite RFF Function Space $\mathcal{H}_{\mathrm{RFF}}$ with respect to the $\mathcal{L}^2(\Omega, \mathbb{R})$ norm.*

*Proof.* Let $g_1$ be an arbitrary function in $\mathcal{H}_f$. Since $g_1$ is a linear combination of the continuous functions in $h(x)$, $g_1$ is a continuous function on a compact set $\mathcal{X}$, i.e., $g_1 \in C(\mathcal{X})$. By Theorem 1, the composite kernel $k_h$ is universal. Thus, its RKHS $\mathcal{H}_{k_h}$ is dense in $C(\mathcal{X})$ under the uniform norm. This means that for any $\epsilon > 0$, there exists a function $f_{k_h} \in \mathcal{H}_{k_h}$ such that:

$$\|g_1 - f_{k_h}\|_\infty = \sup_{x \in \mathcal{X}} |g_1(x) - f_{k_h}(x)| < \frac{\epsilon}{2}.$$

For any probability measure $\rho$, the $L^2(\rho)$ norm is bounded by the $L_\infty$ norm, that is:

$$\|g_1 - f_{k_h}\|_{L^2(\rho)}^2 = \int_\mathcal{X} |g_1(x) - f_{k_h}(x)|^2 d\rho(x)$$

$$\leq \int_\mathcal{X} \left( \sup_{z \in \mathcal{X}} |g_1(z) - f_{k_h}(z)| \right)^2 d\rho(x)$$

$$= \|g_1 - f_{k_h}\|_\infty^2 \int_\mathcal{X} d\rho(x) = \|g_1 - f_{k_h}\|_\infty^2.$$

Thus we have $\|g_1 - f_{k_h}\|_{L^2(\rho)} \leq \|g_1 - f_{k_h}\|_\infty < \frac{\epsilon}{2}$.

From Theorem 3, the composite RFF space $\mathcal{H}_{\mathrm{RFF}}$ is dense in the RKHS $\mathcal{H}_{k_h}$ under the $L^2(\rho)$ norm. Therefore, for our function $f_{k_h}$ from the previous step, there exists a function $f_{\mathrm{RFF}} \in \mathcal{H}_{\mathrm{RFF}}$ such that:

$$\|f_{k_h} - f_{\mathrm{RFF}}\|_{L^2(\rho)} < \frac{\epsilon}{2}.$$

Combining the results using the triangle inequality for the $L^2$ norm:

$$\|g_1 - f_{\mathrm{RFF}}\|_{L^2(\rho)} \leq \|g_1 - f_{k_h}\|_{L^2(\rho)} + \|f_{k_h} - f_{\mathrm{RFF}}\|_{L^2(\rho)} < \frac{\epsilon}{2} + \frac{\epsilon}{2} = \epsilon.$$

Since for any $g_1 \in \mathcal{H}_f$ and any $\epsilon > 0$, we have found an element $f_{\mathrm{RFF}} \in \mathcal{H}_{\mathrm{RFF}}$ that is $\epsilon$-close in the $L^2$ norm, we have proven that $\mathcal{H}_f \subseteq \overline{\mathcal{H}_{\mathrm{RFF}}}$. $\square$

### D.2 Proof of Theorem 2 (Projection Error Comparison)

We have shown in Lemma 2 that the premise $\mathcal{H}_f \subseteq \overline{\mathcal{H}_{\mathrm{RFF}}}$ holds. The proof now proceeds as follows.

For any function $g \in L^2$, define its $L^2$ approximation error with respect to $f$ as $E(g) := \|f - g\|_{L^2}$. The function $E : L^2 \to \mathbb{R}$ is continuous, which follows from the reverse triangle inequality, establishing that the norm is a continuous function.

Let $g_1$ be an arbitrary element in $\mathcal{H}_f$. Since $\mathcal{H}_f \subseteq \overline{\mathcal{H}_{\mathrm{RFF}}}$, by the definition of closure, there exists a sequence of functions $\{g_2^{(n)}\}_{n=1}^\infty$ in $\mathcal{H}_{\mathrm{RFF}}$ such that $g_2^{(n)} \to g_1$ in the $L^2$ sense, that is,

$$\lim_{n \to \infty} \|g_2^{(n)} - g_1\|_{L^2} = 0.$$

Because the error function $E(\cdot)$ is continuous, we can interchange the function with the limit:

$$\lim_{n \to \infty} E(g_2^{(n)}) = E \left( \lim_{n \to \infty} g_2^{(n)} \right) = E(g_1).$$

For each $n$, $g_2^{(n)}$ is an element of $\mathcal{H}_{\mathrm{RFF}}$, so its error $E(g_2^{(n)})$ must be greater than or equal to the infimum of errors over $\mathcal{H}_{\mathrm{RFF}}$:

$$\inf_{g \in \mathcal{H}_{\mathrm{RFF}}} E(g) \leq E(g_2^{(n)}).$$

This inequality holds for all $n$. Taking the limit as $n \to \infty$ on both sides yields

$$\inf_{g \in \mathcal{H}_{\mathrm{RFF}}} E(g) \leq \lim_{n \to \infty} E(g_2^{(n)}).$$

Substituting the continuity result gives

$$\inf_{g \in \mathcal{H}_{\text{RFF}}} E(g) \le E(g_1).$$

The inequality holds for any arbitrary $g_1 \in \mathcal{H}_f$. This implies that $\inf_{g \in \mathcal{H}_{\text{RFF}}} E(g)$ is a lower bound for the set of values $\{E(g) \mid g \in \mathcal{H}_f\}$. By the definition of an infimum (greatest lower bound), this value must be less than or equal to the infimum of the set:

$$\inf_{g \in \mathcal{H}_{\text{RFF}}} E(g) \le \inf_{g \in \mathcal{H}_f} E(g).$$

### D.2.1 PROOF OF THE UNIVERSAL APPROXIMATION COROLLARY

The following inequality holds:

$$\|u - u_{\omega,\theta}\|^2 \le \|u - \tilde{u}_{\omega,W}\|^2 + \|\tilde{u}_{\omega,W} - u_{\omega,\theta}\|^2.$$

The universal approximation theorem by Hornik (1991) guarantees that for any $\varepsilon > 0$, there exists $p$ and $\omega_*, W_*$ such that $\|u - \tilde{u}_{\omega_*,W}\|^2 \le \varepsilon/2$. Theorem 2 of the main paper ensures that there exists $D$ and $\theta_*$ such that $\|\tilde{u}_{\omega_*,W_*} - u_{\omega_*,\theta_*}\|^2 \le \varepsilon/2$. Consequently, the norm between $u$ and $u_{\omega_*,\theta_*}$ is smaller than $\varepsilon$ and that concludes the proof.

## E IFeF-PINN HYPERPARAMETERS

This section details the hyperparameters used by the proposed IFeF-PINN method in each experiment (Table 1). Here, $D$ denotes the number of Fourier-enhanced features; $\sigma$ is the standard deviation of the sampled frequencies in random Fourier features (RFF); $\gamma$ is the regularization parameter defined in Eq. equation 4; and Pre-training indicates a warm-up stage where a vanilla PINN is trained for several thousand epochs to provide a good initialization for basis selection in IFeF-PINN.

As discussed in Wang et al. (2021), the selection of $\sigma$ should align with the target function's frequency content. However, we fix $\sigma = 1$ across all cases in our experiments and obtain accurate approximations. In future work, a more detailed analysis of $\sigma$ will be presented.

| Problem | Pre-training | $D$ | $\sigma$ | $\lambda_{\text{LL}}$ | $\gamma$ |
|---|---|---|---|---|---|
| 2D Helmholtz ($a_1 = 1, a_2 = 4$) | No | 800 | 1 | 1e-2 | 1e-6 |
| 2D Helmholtz ($a_1 = a_2 = 100$) | No | 2400 | 1 | 1e-7 | 1e-4 |
| 1D Convection ($\beta = 50$) | Yes | 800 | 1 | 1e-2 | 1e-7 |
| 1D Convection ($\beta = 200$) | Yes | 1600 | 1 | 1e-2 | le-4/1e-7 |
| Viscous Burgers ($\nu = \frac{0.01}{\pi}$) | Yes | 800 | 1 | 1e-1 | 0 |
| Convection-Diffusion ($k_{\text{low}} = 4\pi, k_{\text{high}} = 60\pi$) | Yes | 800 | 1 | 1e-2 | 1e-7 |

Table 1: Hyperparameters setting for IFeF-PINN under each experiment

## F EXPERIMENT SETUP

### F.1 PDEs SETUP

In this section, we provide detailed PDE settings used as our benchmarks.

**2D Helmholtz Equation.** The Helmholtz equation is an elliptic PDE that commonly arises in the study of wave propagation, acoustics, and electromagnetic fields. We consider the 2D Helmholtz equation as follows:

$$\begin{aligned} \nabla^2 u + u &= f, \quad (x,y) \in \Omega, \\ u(x,y) &= 0, \quad (x,y) \in \partial\Omega, \end{aligned} \tag{14}$$

corresponding to a source term

$$f(x,y) = -\pi^2 \left( a_1^2 \sin(a_1\pi x)\sin(a_2\pi y) - a_2^2 \sin(a_1\pi x)\sin(a_2\pi y) \right) + \sin(a_1\pi x)\sin(a_2\pi y).$$

The parameters $a_1$ and $a_2$ define the frequency of the analytic solution $u(x,y) = \sin(a_1\pi x)\sin(a_2\pi y)$. We will investigate the following different frequency cases:

- low-frequency: $a_1 = 1, a_2 = 4, \Omega = [-1, 1] \times [-1, 1]$.
- high-frequency: $a_1 = 100, a_2 = 100, \Omega = [0, 0.2] \times [0, 0.2]$.

The high-frequency case uses a reduced domain to maintain computational tractability. By the Nyquist-Shannon sampling criterion Shannon (2006), resolving such high-frequency oscillations over a larger domain would require prohibitively dense collocation. Despite the smaller domain, the configuration covers $10 \times 10$ wavelengths, capturing the extreme oscillatory behavior.

**1D Convection Equation.** The Convection equation is a hyperbolic PDE that describes the movement of a substance through fluids. We consider the periodic boundary conditions system as follows:

$$
\begin{aligned}
&\frac{\partial u}{\partial t} + \beta \frac{\partial u}{\partial x} = 0, \quad (t, x) \in [0, 1] \times [0, 2\pi], \\
&u(x, 0) = \sin x, \\
&u(0, t) = u(2\pi, t).
\end{aligned}
\tag{15}
$$

The closed-form solution is $u(x, t) = \sin(x - \beta t)$. We consider a low-frequency case $\beta = 50$ and a high-frequency case $\beta = 200$ on the same domain.

**1D Convection-Diffusion Equation.** The Convection–Diffusion equation is a parabolic PDE that models the combined effects of transport by fluid motion and spreading due to diffusion. We consider the multi-scale system with periodic boundary conditions as follows:

$$
\begin{aligned}
&\frac{\partial u}{\partial t} + c \frac{\partial u}{\partial x} = d \frac{\partial^2 u}{\partial x^2}, \quad (t, x) \in [0, 1] \times [0, 1], \\
&u(t, 0) = u(t, 1), \\
&\frac{\partial u}{\partial x}(t, 0) = \frac{\partial u}{\partial x}(t, 1), \\
&u(0, x) = A_{\text{low}} \sin(k_{\text{low}} x) + A_{\text{high}} \sin(k_{\text{high}} x).
\end{aligned}
\tag{16}
$$

The analytic multi-scale solution is

$$
u(t, x) = A_{\text{low}} e^{-d k_{\text{low}}^2 t} \sin\left(k_{\text{low}}(x - ct)\right) + A_{\text{high}} e^{-d k_{\text{high}}^2 t} \sin\left(k_{\text{high}}(x - ct)\right).
$$

To set a multi-scale problem consists of both low- and high-frequency components, the parameters are chosen as follows:

$$
c = 1, d = 0.00005, A_{\text{low}} = 1, A_{\text{high}} = 0.1, k_{\text{low}} = 4\pi, k_{\text{high}} = 60\pi.
$$

**Viscous Burgers' Equation.** The Viscous Burgers' equation is a nonlinear parabolic PDE that models fluid motion by combining convection and diffusion effects. We consider the nonlinear system as follows:

$$
\begin{aligned}
&\frac{\partial u}{\partial t} + u \frac{\partial u}{\partial x} = \nu \frac{\partial^2 u}{\partial x^2}, \quad (t, x) \in [0, 1] \times [-1, 1], \\
&u(0, x) = -\sin(\pi x), \\
&u(t, -1) = u(t, 1) = 0,
\end{aligned}
\tag{17}
$$

where $\nu = \frac{0.01}{\pi}$.

### F.2 SPECTRUM ANALYSIS SETUP

Given frequencies $\kappa = \{f_i\}_{i=1}^{i=10} = \{1, 2, 5, 10, 30, 40, 50, 60, 70, 80\}$, where all amplitudes are chosen as $A_i = 1$, we consider the Convection equation in 15 with $\beta = 1$ and initial condition as follows:

$$
u(x, 0) = \sum_{i=1}^{10} A_i \sin(2\pi f_i \, x).
$$

The corresponding analytic solution is then given by

$$
u(x, t) = \sum_{i=1}^{10} A_i \sin\left(2\pi f_i \, (x - t)\right).
$$

The problem domain is defined as $(t, x) \in [0, 1] \times [0, 1]$. Our objective is to evaluate and compare the ability of Vanilla PINNs and Fourier-enhanced Features to capture all frequency components at $t = 0$.

For the neural network architecture, we employ an 8-layer fully connected network with $\tanh$ activation functions and 64 neurons per layer. The training data consist of 201 uniformly sampled points along the two spatial boundaries ($x = 0$ and $x = 1$) and at the final time ($t = 1$). In addition, $201 \times 201$ collocation points are uniformly sampled within the interior domain to enforce the physics constraints.

To further design this experiment as an ablation study and demonstrate that the incorporation of Fourier-enhanced Features for basis extension is a necessary component of our proposed IFeF-PINN, we discard the iterative training procedure and retain only the basis extension step. Specifically, we first train the Vanilla PINN for 40,000 epochs using the Adam optimizer with a learning rate of $10^{-3}$. We then extend the basis with varying numbers of Fourier-enhanced Features, $D_j \in \{400, 800, 1600, 2400, 3200, 4000\}$, and then solve the lower-level problem defined in Equation 6 of the main paper. Moreover, to ensure a more rigorous analysis, we impose the relation $B_{D_i} \subset B_{D_j}$ whenever $D_j > D_i$, so that the RFF mapping matrices are nested.

### F.3 MODEL SETUP

This section details the model setup for all baselines. Unless otherwise specified, we use a multi-layer perceptron (MLP) whose depth and width are determined by the experimental setting. For the 2D Helmholtz case ($a_1 = 1$, $a_2 = 4$), we follow the network structure in Barreau & Shen (2025); for the viscous Burgers' equation, we follow Raissi et al. (2019). For PINNsformer (Zhao et al., 2024) and PIG (Kang et al., 2025), since they both have special network architectures, we adopt the original architecture. All experiments use the $\tanh$ activation function and the Adam optimizer with a learning rate of $10^{-3}$ for the network parameters. For IFeF-PD, we adopt the Primal-Dual weight balancing strategy proposed in Barreau & Shen (2025), and optimize the dynamic physics weight with the same setting for Adam at a learning rate of $10^{-4}$. The network architectures used in each experiment are summarized in Table 2.

| Problem | Hidden layers | Hidden width |
|---|---|---|
| 2D Helmholtz ($a_1 = 1, a_2 = 4$) | 3 | [50,50,20] |
| 2D Helmholtz ($a_1 = a_2 = 100$) | 6 | 64 |
| 1D Convection ($\beta = 50$) | 6 | 64 |
| 1D Convection ($\beta = 200$) | 6 | 64 |
| Viscous Burgers ($\nu = \frac{0.01}{\pi}$) | 8 | 20 |
| Convection-Diffusion ($k_{\text{low}} = 4\pi, k_{\text{high}} = 60\pi$) | 6 | 64 |

Table 2: Network architecture for all problems.

### F.4 DATASET SETUP

In this section, we detail the dataset setup for each equation and experiment. For the 1D Convection equation, 2D Helmholtz equation (low-frequency), and Convection-Diffusion equation, we follow the setting and strategy of Zhao et al. (2024). For the Viscous Burgers' Equation, we follow Raissi et al. (2019). The detailed settings are summarized in Table 3.

## G EXPERIMENTAL RESULTS

In this section, we present the true solutions, model predictions, and absolute error maps for all baselines considered in our numerical experiments. Results for the viscous Burgers' equation, the low- and high-frequency convection equations, and the multi-scale convection-diffusion equation are shown in separate figures. For clarity, each figure contains three panels: (i) the true solution, (ii) the model prediction, and (iii) the absolute error on a log10 scale.

| Problem | Sampling | Boundary points | Physics points |
|---|---|---|---|
| 2D Helmholtz ($a_1 = 1, a_2 = 4$) | Uniform | 1000 | $71 \times 71$ |
| 2D Helmholtz ($a_1 = a_2 = 100$) | LHS | 3000 | 23000 |
| 1D Convection ($\beta = 50$) | Uniform | $303^\dagger$ $153^\ddagger$ | $(101 \times 101)^\dagger$ $(51 \times 51)^\ddagger$ |
| 1D Convection ($\beta = 200$) | Uniform | 303 | $101 \times 101$ |
| Viscous Burgers ($\nu = \frac{0.01}{\pi}$) | LHS | 100 | 10000 |
| Convection-Diffusion ($k_{\text{low}} = 4\pi, k_{\text{high}} = 60\pi$) | Uniform | 404 | $101 \times 101$ |

Table 3: Dataset settings for each PDE problem.

Notes: $^\dagger$ Vanilla/NTK/PIG; $^\ddagger$ PINNsformer/IFeF.

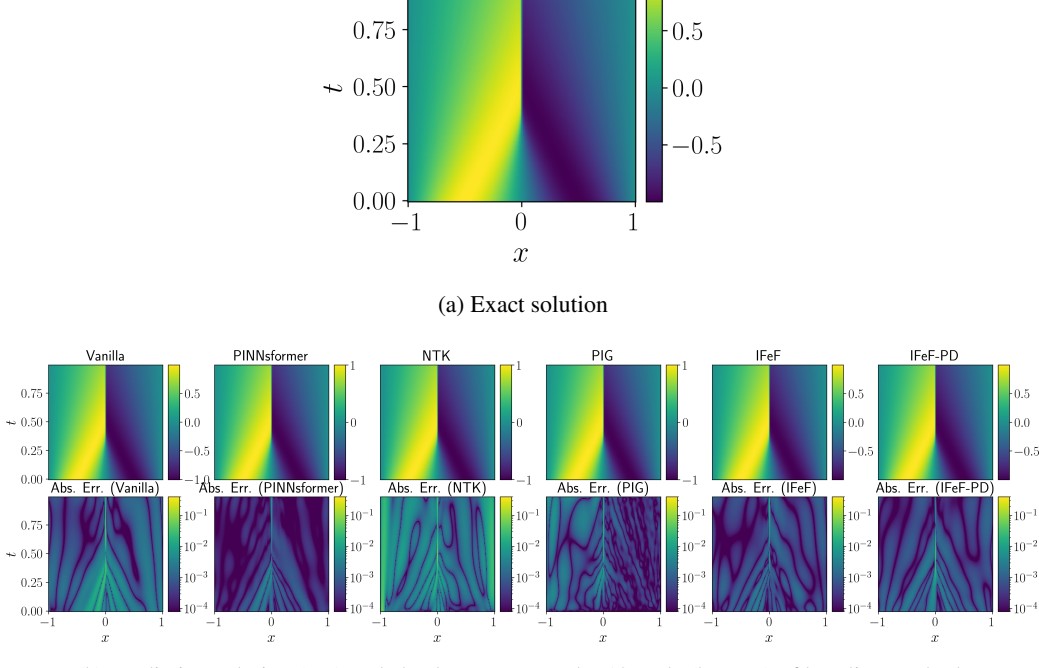

(a) Exact solution

(b) Prediction solution (top) and absolute error on a log10 scale (bottom) of baseline methods

Figure 1: True solution, prediction and absolute error of baseline methods for viscous Burgers' equation

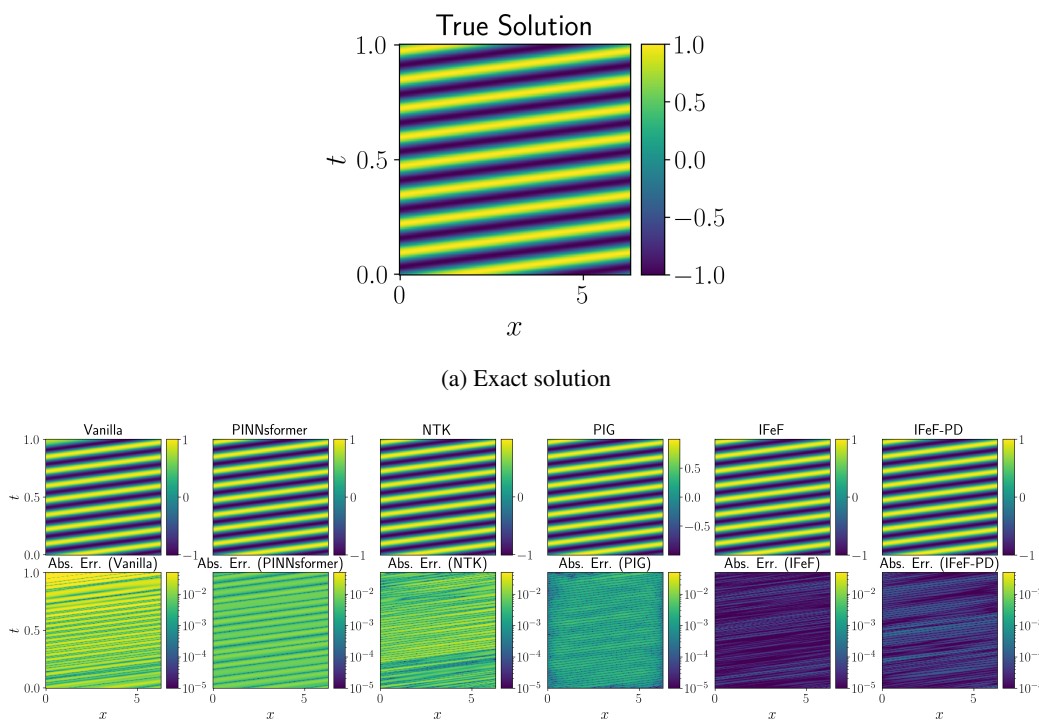

(a) Exact solution

(b) Prediction solution (top) and absolute error on a log10 scale (bottom) of baseline methods

Figure 2: True solution, prediction, and absolute error of baseline methods for low-frequency convection equation

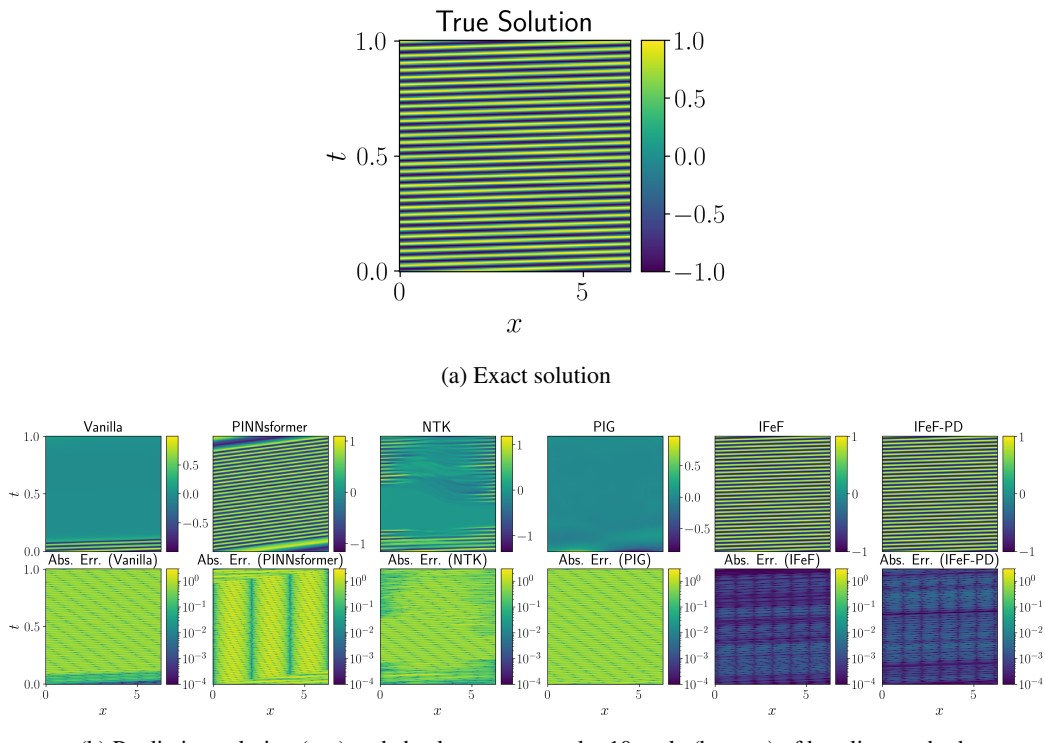

(a) Exact solution

(b) Prediction solution (top) and absolute error on a log10 scale (bottom) of baseline methods

Figure 3: True solution, prediction, and absolute error of baseline methods for high-frequency convection equation

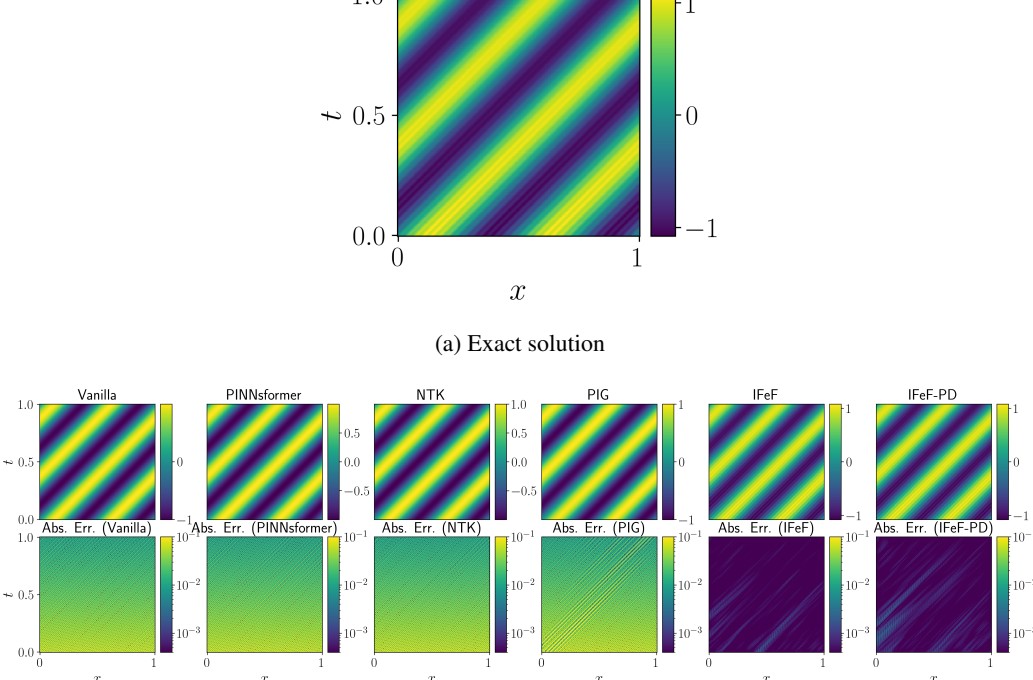

(a) Exact solution

(b) Prediction solution (top) and absolute error on a log10 scale (bottom) of baseline methods

Figure 4: True solution, prediction, and absolute error of baseline methods for multi-scale convection-diffusion equation

The case presented in Figure 4 is particularly interesting. We observe that the analytical solution is the sum of the two frequencies (20 and 60). If we zoom in on the error plot, it is possible to see that for all other methods than IFeF, the high-frequency component is not caught. Despite visually similar plots, the error can be quite large. This phenomenon does not appear with IFeF-PINN.

### G.1 COMPARISON WITH INPUT-SPACE RFF

We compare IFeF-PINN with the Multi-scale Fourier Features (MFF) method proposed by Wang et al. (2021), which applies multiple RFF mappings derived in Equation 4 of the main paper to the input layer of the neural network to mitigate spectral bias.

Table 4 summarizes the relative $L^2$-error and standard deviation across the low-frequency, high-frequency and multi-scale benchmarks. IFeF-PINN consistently outperforms MFF in approximation performance.

| **Baseline** | Convection $(\beta = 50)$ | Convection $(\beta = 200)$ | Convection-Diffusion $(k_{\text{low}} = 4\pi,\ k_{\text{high}} = 60\pi)$ |
|---|---|---|---|
| MFF | $2.14 \times 10^{-2}(4.42 \times 10^{-3})$ | $3.50 \times 10^{-1}(2.27 \times 10^{-1})$ | $5.21 \times 10^{-2}(4.21 \times 10^{-4})$ |
| IFeF | $7.0 \times 10^{-5}(1.6 \times 10^{-3})$ | $2.7 \times 10^{-3}(1.0 \times 10^{-3})$ | $9.0 \times 10^{-4}(3.0 \times 10^{-4})$ |
| IFeF-PD | $9.0 \times 10^{-5}(5.0 \times 10^{-4})$ | $2.5 \times 10^{-3}(5.0 \times 10^{-4})$ | $1.0 \times 10^{-3}(2.0 \times 10^{-4})$ |

Table 4: Average relative $L^2$-error with corresponding standard deviation across 3 benchmarks for IFeF-PINN, IFeF-PD and MFF.

### G.2 COMPARISON WITH COMPUTATIONAL COST

We provide a comparison of the computational costs for the 5 linear benchmarks among IFeF-PINN, vanilla PINNs, and the SOTA baseline PIG proposed by Kang et al. (2025). Tables 5 and 6

summarize the average training time per epoch, total training time, and memory usage for the three methods. To better analyze the computational cost in IFeF-PINN, we decompose the per-epoch training time into the upper-level and lower-level components.

| IFeF-PINN | Per upper (s) | Per lower (s) | Total time (s) | Memory (GB) |
|---|---|---|---|---|
| 2D Helmholtz ($a_1 = 1, a_2 = 4$) | 0.015 | 0.003 | 448 | 1.41 |
| 2D Helmholtz ($a_1 = a_2 = 100$) | 0.154 | 0.042 | 1960 | 18.5 |
| 1D Convection ($\beta = 50$) | 0.024 | 0.003 | 108 | 4.80 |
| 1D Convection ($\beta = 200$) | 0.051 | 0.010 | 610 | 5.85 |
| Convection-Diffusion ($k_{\text{low}} = 4\pi, k_{\text{high}} = 60\pi$) | 0.052 | 0.003 | 110 | 5.26 |

Table 5: Average training time per epoch for upper- and lower-level, total training time and memory usage for IFeF-PINN among 5 linear benchmarks.

| | Vanilla PINN | | | PIG | | |
|---|---|---|---|---|---|---|
| **Problem** | Per epoch (s) | Total (s) | Memory (GB) | Per epoch (s) | Total (s) | Memory (GB) |
| 2D Helmholtz ($a_1 = 1, a_2 = 4$) | 0.003 | 116 | 0.2 | 0.62 | 248 | 6.8 |
| 2D Helmholtz ($a_1 = a_2 = 100$) | 0.006 | - | 0.55 | 1.70 | - | 20.6 |
| 1D Convection ($\beta = 50$) | 0.002 | 18 | 0.10 | 1.13 | 565 | 5.8 |
| 1D Convection ($\beta = 200$) | 0.002 | - | 0.10 | 1.63 | - | 14.7 |
| Convection-Diffusion ($k_{\text{low}} = 4\pi, k_{\text{high}} = 60\pi$) | 0.003 | 27 | 0.19 | 0.85 | 43 | 9.8 |

Table 6: Average training time per epoch, total training time and memory usage for Vanilla PINNs and PIG among 5 linear benchmarks. A dash '-' denotes that the method failed to achieve a meaningful approximation for the corresponding equation and is therefore excluded from the total training time.

The results demonstrate that vanilla PINN achieves the fastest training but the poorest accuracy. PIG with its default L-BFGS optimizer converges in fewer epochs but incurs the highest memory cost due to the evaluation of numerous learnable Gaussian bases at collocation points. IFeF-PINN demonstrates lower memory usage than PIG while maintaining acceptable training time. Notably, IFeF-PINN's training is dominated by upper-level basis learning, while the lower-level QP solving is highly efficient.