# OpenReview forum: "Iterative Training of Physics-Informed Neural Networks with Fourier-enhanced Features"
_ICLR.cc/2026/Conference — ICLR 2026 Poster_

### Official Review · Reviewer_F9RY · 2025-10-19

**Soundness:** 2
**Presentation:** 3
**Contribution:** 2
**Rating:** 6
**Confidence:** 3

**Summary:**

This paper proposes IFeF-PINN, an algorithm for iterative training of PINNs with Fourier-enhanced features. The core of IFeF-PINN is to enrich the latent space using high-frequency components through Random Fourier Features (RFF).  This work provides detailed theoretical derivations for several key conclusions. Experiments also show that IFeF-PINN outperforms other SOTA methods.

**Strengths:**

- The paper is well-written and easy to follow.
- The claims of this paper are substantiated by multiple theoretical derivations.
- The effectiveness of IFeF-PINN is validated on various problems.

**Weaknesses:**

- The two-stage training process is a key claim of this work, but it lacks sufficient justification and experimental evidence to demonstrate its necessity. The reviewer acknowledges the importance of decoupled representation and downstream tasks (e.g., large-scale pretraining). However, as the significance of this pipeline in the scope of PINNs is not yet established, the authors should provide a more thorough discussion. An ablation study comparing against an end-to-end training approach using RFF basis extension would help clarify this issue.
- The paper lacks comparison with a basic baseline: the method described in Equation 4. The reviewer disagrees with the authors' claim in Line 155 ("when the input is of low dimension, such as for PINNs, this could limit its asymptotic approximation capabilities"). The method in Equation 4 has been successfully validated in NeRF (NeRF: Representing Scenes as Neural Radiance Fields for View Synthesis), which also utilizes low-dimensional inputs (3D location and 2D viewing direction).
- IFeF-PINN involves multiple training phases, potentially requiring more training steps. The authors mention in Line 344 that baseline methods are trained/evaluated with their default settings. The reviewer questions the fairness of comparisons and suggests that ensuring all methods use the same total number of training steps would provide a more equitable evaluation.
- The impact of the number of Fourier-enhanced features (D) on performance is unclear. Is there a correlation between the value of D and downstream task performance?

**Questions:**

Please refer to the weaknesses. The reviewer would consider raising the score if the first two weaknesses are adequately addressed.

---

> ### Author Response · Authors · 2025-11-20
> **W1**
>
> We are grateful for the detailed feedback and the specific suggestions for strengthening our work. We will address each point regarding the justification of two-stage training, comparisons with the end-to-end training, and the baseline in Equation 4, fair evaluation, and the impact of feature number $D$.
>
> ## **W1. Justification and End-to-End ablation experimental evidence for two-stage training**
>
> We conduct a new ablation study to demonstrate the necessity of the two-stage training. The design for the end-to-end training approach using RFF basis extension is as follows:
>
> To keep the RFF mapping to the last hidden layer output, we still have $\psi_D(x) = \gamma_D\left(h_{\omega}(x)\right) = \frac{1}{\sqrt{D}} \begin{bmatrix} \cos(2\pi B_D h_\omega(x)) ,
>             \sin(2\pi B_D h_\omega(x)) \end{bmatrix}$ and approximate the PDE with the extended basis as:
> \begin{equation*}
>     u_{\omega,\theta}(x) = \psi_D(x)^\top \theta, \quad x \in \Omega.
> \end{equation*}
>
>
> Then we still have the overall loss function by bringing this approximation into the loss function of PINNs
> \begin{equation*}
> \mathfrak{L}(\omega,\theta) = \\frac{1}{N_u} \\sum_{i=1}^{N_u} \| g(x_u^i) - \mathfrak{B}[u_{\omega,\theta}]{x_u}^{i}  \|^2 + \\frac{\lambda}{N_f} \\sum_{i=1}^{N_f} \|\mathfrak{F}[u_{\omega,\theta}]x_f^i \|^2.
> \end{equation*}
>
> In the IFeF-PINN two-stage training approach, $\theta$ is always optimal with respect to the current features $\psi_D(x)$ in the lower-level problem, then the network parameter $\omega$ is updated based on $\theta^\star(\omega)$. In end-to-end training, both $\theta$ and $\omega$ become learnable parameters of a neural network. They are jointly and simultaneously optimized by the Adam optimizer. $\theta$ is initialized randomly and is not guaranteed to be optimal at any given iteration.
>
> We maintain all datasets and hyperparameter settings, and tested this ablation study with low-frequency, high-frequency Helmholtz equations and Burgers equations. The result is shown below:
>
> In the IFeF-PINN two-stage training approach, $\theta$ is always optimal with respect to the current features $\psi_D(x)$ in the lower-level problem, then the network parameter $\omega$ is updated based on $\theta^\star(\omega)$. In end-to-end training, both $\theta$ and $\omega$ become learnable parameters of a neural network. They are jointly and simultaneously optimized by the Adam optimizer. $\theta$ is initialized randomly and is not guaranteed to be optimal at any given iteration.
>
> We maintain all datasets and hyperparameter settings, and tested this ablation study with low-frequency, high-frequency Helmholtz equations and Burgers equations. The result is shown below:
>
> | **Ablation**   | Helmholtz $(a_1=1, a_2=4)$               | Helmholtz $(a_1=a_2=100)$                | Viscous Burgers$(\nu = 0.01/\pi)$      |
> | -------------- | ---------------------------------------- | ---------------------------------------- | ---------------------------------------- |
> | **End-to-End** | $8.81 \times 10^{-3}(6.0\times10^{-4})$ | –                                        | $4.93 \times 10^{-3}(8.5\times10^{-4})$ |
> | **IFeF**       |$3.2 \times 10^{-4}(2.6\times10^{-4})$  | $1.56 \times 10^{-2}(5.5\times10^{-3})$ | $2.41 \times 10^{-3}(1.1\times10^{-3})$ |
> | **IFeF-PD**    | $4.0 \times 10^{-5}(2.0\times10^{-5})$  | $9.2 \times 10^{-3}(3.1\times10^{-3})$  | $3.27 \times 10^{-3}(4.1\times10^{-4})$ |
>
> This valuable ablation study reveals the benefits and drawbacks of our IFeF-PINN training approach. For linear PDEs (Helmholtz equations), IFeF-PINN achieves much lower error than end-to-end training because the lower-level problem has a closed-form global optimizer, ensuring that $\theta$ is always optimal given the current features $\omega$. This decoupling allows the upper-level optimization to focus solely on learning features.
> However, for nonlinear PDEs (Burgers equation), the improvement is modest, as the lower-level optimization becomes non-convex and can be easily stuck in a local minimizer, diminishing the advantage of decoupling.
>
> This observation highlights the superior performance of IFeF-PINN for linear PDEs, while maintaining viability for nonlinear problems. We thank the reviewer for this insightful comment and will include this comprehensive End-to-End ablation study and discussion in the revised manuscript to clarify the scope and limitations of our approach.

---

> > ### Author Response · Authors · 2025-11-20
> > **Supplement to W1**
> >
> > Additionally, we would like to provide a theoretical motivation for decoupling the network into a two-stage training and using RFF mapping on the last hidden layer of the network. By Theorem 3 in the Appendix and the proof derived in [1], we show that $k_h(x, x') := k(h(x), h(x')) \approx \psi^\top_D(x) \psi_D(x)$ is a universal kernel and is also a shift-invariant kernel under $h_{\omega}$. Consider the loss function of the lower-level problem
> >
> > \begin{equation*}
> >     \mathfrak{L}(u_{\omega,\theta})
> >     = \frac{1}{N_u} \bigl\| B_u(\omega) \theta - G_u \bigr\|^2
> >     + \frac{\lambda}{N_f}\bigl\| R_f(\omega) \theta - F_f \bigr\|^2 + \gamma \| \theta \|^2.
> > \end{equation*}
> >
> > Then we have the normal equation for $\theta$ as
> > \begin{equation*}
> >     (\frac{1}{N_u} B_u^\top B_u + \frac{\lambda}{N_f} R_f\top R_f + \gamma I)\theta = \frac{1}{N_u} B_u^\top G_u + \frac{\lambda}{N_f} R_f\top F_f
> > \end{equation*}
> > Then we can rewrite the lower-level problem using kernel representation to convert the lower-level problem to a kernel ridge regression problem as
> > \begin{equation*}
> > \Psi \coloneqq \begin{pmatrix} \sqrt{\frac{1}{N_u}}B_u ,   \sqrt{\frac{\lambda}{N_f}}R_f \end{pmatrix}, \quad
> > y \coloneqq \begin{pmatrix} \sqrt{\frac{1}{N_u}}G_u ,   \sqrt{\frac{\lambda}{N_f}}F_f \end{pmatrix},
> > \end{equation*}
> > This is precisely the dual form of kernel ridge regression with a positive-definite Gram matrix $K = \Psi \Psi^T \in \mathbb{R}^{(N_u+N_f)\times (N_u+N_f)}$, with the eigenvalue and spectral properties of the kernel directly related to $k_h$. Then using the similar proof path as shown in~\cite{tancik2020fourier}, we can tune $\sigma$ to control the frequency spectrum of the kernel, thereby mitigating spectral bias. This kernel perspective justifies the IFeF-PINN structure: by decoupling the problem into two stages and explicitly employing RFF mapping, we construct a universal shift-invariant kernel $k_h$ whose spectral properties can be controlled through $\sigma$.
> >
> > The above kernel ridge regression perspective theoretically addresses the reviewer's concern regarding the necessity of the decoupled representation and two-stage training process. We will add a new subsection to elaborate on this theoretical justification and clarify the essential role of the two-stage and RFF mapping design in controlling spectral properties.
> >
> > [1] Tancik, Matthew, Pratul Srinivasan, Ben Mildenhall, Sara Fridovich-Keil, Nithin Raghavan, Utkarsh Singhal, Ravi Ramamoorthi, Jonathan Barron, and Ren Ng.
> > "Fourier features let networks learn high-frequency functions in low-dimensional domains."
> > Advances in Neural Information Processing Systems 33 (2020): 7537–7547.

---

> ### Author Response · Authors · 2025-11-20
> **W2**
>
> ## **W2. The paper lacks comparison with a basic baseline: the method described in Equation 4. The reviewer disagrees with the authors' claim in Line 155.**
>
> We agree that the method described in Equation 4, applying RFF mapping directly to the input layer should be included as a baseline. This approach, known as Multi-scale Fourier Features (MFF) proposed by [1], employs RFF mapping to address spectral bias in PINNs.
>
> We conducted additional experiments using MFF with its default hyperparameter settings on three representative problems: low-frequency Convection equation, high-frequency Convection equation, and multi-scale Convection-Diffusion equation. The relative $L^2$ errors averaged over five runs are presented below:
> | **Baseline** | Convection ($\beta=50$)                    | Convection ($\beta=200$)                   | Convection–Diffusion ($k_\text{low}=4, k_\text{high}=60$) |
> | ------------ | ------------------------------------------ | ------------------------------------------ | --------------------------------------------------------- |
> | **MFF**      | $2.14 \times 10^{-2}(4.42\times 10^{-3})$ | $3.50 \times 10^{-1}(2.27\times 10^{-1})$ | $5.21 \times 10^{-2}(4.21\times 10^{-4})$               |
> | **IFeF**     | $7.0 \times 10^{-5}(1.6\times10^{-3})$    | $2.7 \times 10^{-3}(1.0\times10^{-3})$    | $9.0 \times 10^{-4}(3.0\times10^{-4})$                   |
> | **IFeF-PD**  | $9.0 \times 10^{-5}(5.0\times10^{-4})$   | $2.5 \times 10^{-3}(5.0\times10^{-4})$    | $1.0 \times 10^{-3}(2.0\times10^{-4})$                   |
>
> The results for MFF demonstrate its ability to mitigate spectral bias, particularly for the high-frequency convection equation (with the best relative $L^2$ error of $5.9\times10^{-2}$ among all runs). However, the multi-scale convection-diffusion results reveal that it still struggles to capture both low-frequency and high-frequency oscillatory components simultaneously, and IFeF-PINN shows better performance overall.
>
> Our statement in Line 155 is too absolute and does not accurately reflect the RFF mapping to low-dimensional inputs. NeRF demonstrates that input-layer RFF (positional encoding) can be highly effective even with low-dimensional inputs (5D in their case). We acknowledge this imprecise and absolute characterization and will revise this statement in the manuscript.
>
> However, there are some differences between the NeRF and PINNs architecture.
> Firstly, NeRF learns from dense pixel-level observations across thousands of images. In contrast, PINNs only have a small dataset for measurements and rely heavily on physics residuals from the PDEs. Secondly, our key motivation for applying RFF to last hidden layer output $h_\omega(x)$ rather than directly to the input layer is to create a richer space of basis functions for the lower-level problem.
>
> [1] Wang, Sifan, Hanwen Wang, and Paris Perdikaris.
> "On the eigenvector bias of Fourier feature networks: From regression to solving multi-scale PDEs with physics-informed neural networks."
> Computer Methods in Applied Mechanics and Engineering 384 (2021): 113938.

---

> ### Author Response · Authors · 2025-11-20
> **W3**
>
> ## **W3.Fair comparisons with all baselines**
>
> We would like to clarify that "default settings" in Line 344 refers primarily to hyperparameter settings being kept consistent with the original publications to ensure each baseline operates under its optimal conditions. For fairness, all methods in our experiments use consistent hyperparameters within each experiment and are trained until convergence rather than stopping at a fixed epoch count. Using the same total number of training steps is not fair because some baselines, such as PINNsformer, use a Transformer-based network architecture. For this method, the number of training epochs is relatively small but the runtime for each epoch is much higher than the MLP-based network architecture. Therefore, a fair and detailed comparison should be made in terms of total training time and memory usage.
>
> To demonstrate the fair comparison of the paper, we show the comparison of computation time and memory usage among IFeF-PINN, Vanilla PINN and the SOTA baseline PIG in some benchmark experiments.
>
> ### **IFeF-PINN**
>
> | Problem | Per upper-level (s) | Per lower-level (s) | Training time (s) | Memory (GB) |
> |--------|----------------------|----------------------|-------------------|-------------|
> | 2D Helmholtz ($a_1=1, a_2=4$) | 0.015 | 0.003 | 448 | 1.41 |
> | 2D Helmholtz ($a_1=a_2=100$)  | 0.154 | 0.042 | 1960 | 18.5 |
> | 1D Convection ($\beta=50$)    | 0.024 | 0.003 | 108 | 4.80 |
> | 1D Convection ($\beta=200$)   | 0.051 | 0.010 | 610 | 5.85 |
> | Convection–Diffusion ($k_{\text{low}}=4\pi$, $k_{\text{high}}=60\pi$) | 0.052 | 0.003 | 110 | 5.26 |
>
> ### **Vanilla**
>
> | Problem | Per epoch (s) | Training time (s) | Memory (GB) |
> |--------|----------------|-------------------|-------------|
> | 2D Helmholtz ($a_1=1, a_2=4$) | 0.003 | 116 | 0.2 |
> | 2D Helmholtz ($a_1=a_2=100$)  | 0.006 | -   | 0.55 |
> | 1D Convection ($\beta=50$)    | 0.002 | 18  | 0.10 |
> | 1D Convection ($\beta=200$)   | 0.002 | -   | 0.10 |
> | Convection–Diffusion ($k_{\text{low}}=4\pi$, $k_{\text{high}}=60\pi$) | 0.003 | 27 | 0.19 |
>
> ### **PIG**
>
> | Problem | Per epoch (s) | Training time (s) | Memory (GB) |
> |--------|----------------|-------------------|-------------|
> | 2D Helmholtz ($a_1=1, a_2=4$) | 0.62 | 248 | 6.8 |
> | 2D Helmholtz ($a_1=a_2=100$)  | 1.70 | -   | 20.6 |
> | 1D Convection ($\beta=50$)    | 1.13 | 565 | 5.8 |
> | 1D Convection ($\beta=200$)   | 1.63 | -   | 14.7 |
> | Convection–Diffusion ($k_{\text{low}}=4\pi$, $k_{\text{high}}=60\pi$) | 0.85 | 43 | 9.8 |
>
> A dash '-' denotes that the method failed to achieve a meaningful approximation for the corresponding equation and is therefore excluded from the total training time.
>
> From the results, we observe that Vanilla PINN, which relies only on MLP training without high-dimensional feature mappings, achieves the fastest per-epoch training speed and the shortest training time. However, it also has the worst fitting performance.
>
> For the PIG algorithm, the default optimizer is L-BFGS, resulting in the longest per-epoch training time. However, it requires the least number of epochs to converge, with only around $50$ epochs on the multi-scale Convection-Diffusion equation (though it fails to capture the high-frequency components of the solution). However, because PIG evaluates and differentiates a large number of learnable Gaussian bases for collocation points, it requires substantial memory usage despite using only a lightweight MLP, as the feature learning and embedding process is memory-intensive. In contrast, our IFeF-PINN algorithm demonstrates lower memory usage compared to PIG. While computation time is higher in some cases, it remains acceptable.
>
> Additionally, we observe from the training process visualization that IFeF-PINN typically achieves the error level for most baselines within relatively few epochs and short training time, with remaining training focused on refining small details for the exact solution at already low error levels. We will include relative $L^2$ error vs. training time curves for some benchmarks in the revision to illustrate this rapid initial convergence of IFeF-PINN.

---

> ### Author Response · Authors · 2025-11-20
> **W4**
>
> ## **W4. The impact of the number of Fourier-enhanced features (D) on performance is unclear.**
> We would like to discuss the impact of the feature number $D$ on performance. The number of Fourier-enhanced features $D$ lifts the nominal basis in the lower-level problem to $2D$ dimensions, which enhances the expressiveness of the basis and improves the ability to capture high-frequency signals, as demonstrated in Theorem 2 and our spectrum analysis in Section 6.3.
>
> To follow the reviewer's comment on the number of Fourier-enhanced features ($D$) on performance, we conduct a study on the Helmholtz equation with varying $D$.
>
>
> | **Number of D**                                      | **100**              | **400**              | **800**              | **1200**             | **3000**             | **Vanilla**         |
> | ------------------------------------------ | -------------------- | -------------------- | -------------------- | -------------------- | -------------------- | ------------------- |
> | **Helmholtz** $(a_1=1, a_2=4), \sigma=1.$   | $5.5\times 10^{-4}$  | $2.1\times 10^{-4}$  | $3.2\times 10^{-4}$  | $5.7\times 10^{-4}$  | $4.5\times 10^{-4}$  | $2.9\times 10^{-2}$ |
> |  **Number of D**                                       | **800**              | **1200**             | **1600**             | **2400**             | **3000**             | **Vanilla**         |
> | **Helmholtz** $(a_1=a_2=100), \sigma=1.$    | $7.11\times 10^{-2}$ | $5.40\times 10^{-2}$ | $3.09\times 10^{-2}$ | $1.56\times 10^{-2}$ | $2.22\times 10^{-2}$ | –                   |
>
> This experiment indicates that low-frequency problems typically require fewer features and exhibit lower sensitivity to variations in $D$. In contrast, high-frequency problems generally demand a larger number of features to adequately cover the frequency spectrum of the solution, and consequently show greater sensitivity to changes in $D$.

---

> > ### Comment · Reviewer_F9RY · 2025-11-20
> >
> > This is a supplementary comment regarding the experimental results. The authors have provided a comprehensive response regarding the impact of the number of Fourier-enhanced features ($D$). The reviewer would like to discuss these results further. Please note that this discussion will **not influence the final assessment** of the paper; it is prompted solely by the intriguing phenomena observed in the experiments.
> > ﻿
> >
> > The results show that increasing $D$ is highly effective for high-frequency problems, which aligns with intuition. However, for low-frequency problems, an excessively large $D$ appears to cause significant performance degradation. The reviewer is curious why the accuracy for low-frequency problems does not simply improve and then saturate (or plateau) as $D$ increases. Do the authors have any potential explanations for this phenomenon?

---

> > > ### Author Response · Authors · 2025-11-21
> > > **Answer for supplementary comment**
> > >
> > > We thank the reviewer for this insightful observation. First, we note that in the low-frequency problem, across the wide range of $D$ from 100 to 3000, the error varies by at most $5.7/2.1 \approx 2.7$, suggesting that the approximation remains relatively stable. Regarding why excessively large $D$ causes performance degradation but does not saturate as $D$ increases, particularly for low-frequency problems, we have the following explanations.
> > >
> > > Firstly, from the rank condition perspective discussed in Appendix B.1, the matrix $Q(\omega)$ is possible to be strictly positive definite when $N_u + N_f \geq 2D$. In the low-frequency Helmholtz experiments, we used $N_u + N_f = 1000 + 71 \times 71 = 6041$ sampling points. When $D = 3000$ ($2D = 6000$), this condition is barely satisfied, and even though we add the regularization term, the matrix $Q(\omega)$ may become ill-conditioned in this over-parameterized basis. Additionally, excessively large $D$ creates a high-dimensional parameter space for the RFF extended basis, making the upper-level neural network easier to reach a local minimum, increasing the difficulty of learning good basis functions. To verify this experimentally, we experiment with $D = 3000$ using a larger dataset of $N_u + N_f = 1000 + 101 \times 101 = 11201$ points, obtaining a relative $L^2$ error of $2.3\times10^{-4}$, which demonstrates that the choice of $D$ should be constrained by the size of dataset.
> > >
> > > Secondly, from the kernel approximation perspective discussed in W1, as $D$ increases from a small value, the RFF extended basis may already sufficiently approximate the kernel at a certain threshold $D$, making it enough for covering the spectral domain of the low-frequency solution. Due to the randomness in RFF features, excessively large $D$ inevitably introduces unnecessarily high-frequency components (even with $\sigma=1$) that act as noise for low-frequency function fitting.

---

> > > > ### Comment · Reviewer_F9RY · 2025-11-21
> > > >
> > > > The reviewer thanks the authors for the additional analyses. All concerns have been satisfactorily addressed, and the reviewer has no more questions.

---

> > ### Author Response · Authors · 2025-11-21
> > **Table correction**
> >
> > We apologize for omitting the $D$ values for the high-frequency case in the table. The corrected table is now shown in the answer. The key finding remains the same: low-frequency problems need fewer features and are less sensitive to $D$, whereas high-frequency problems require more features and are more sensitive to $D$.

---

> ### Comment · Reviewer_F9RY · 2025-11-20
>
> The reviewer appreciates the extensive supplementary experiments and detailed explanations provided by the authors to address weaknesses. All concerns have been satisfactorily addressed. This paper contributes novel insights to the design of PINNs. The reviewer has raised the score to 8.

---

> > ### Author Response · Authors · 2025-11-21
> > **Thank you for your feedback!**
> >
> > We sincerely thank the reviewer for the thorough review and positive feedback! Your insightful suggestions have significantly improved the quality of our paper. We will add the end-to-end ablation study and the MFF baseline results into the revised manuscript to further strengthen this work. Once again, we greatly appreciate your time and effort!

---

### Official Review · Reviewer_JZcR · 2025-10-27

**Soundness:** 3
**Presentation:** 3
**Contribution:** 2
**Rating:** 6
**Confidence:** 3

**Summary:**

To address the spectral bias of PINNs, IFeF-PINN is proposed. It is an algorithm for iterative training of PINNs with Fourier-enhanced features, enriching the latent space using high-frequency components through Random Fourier Features. This creates a two-stage training problem: (i) estimate a feature basis, and (ii) perform regression to determine the coefficients. For an underlying linear model, it is shown that the latter problem is convex and that the iterative training scheme converges. Numerical evaluation on classical benchmark problems shows the superior performance of IFeF-PINN over SOTA algorithms is shown.

**Strengths:**

* (Clarity) The paper is easy to follow. Contributions are well clarified and compared with other studies addressing the spectral bias.
* (Technical contribution) The proposed methods effectively combines PINNs and RFFs in a novel way, addressing the limited approximation ability of RFFs in low dimensions.
* (Broader impact & Technical contribution) An application to general neural architectures is straightforward, with some careful treatments described in Remark 2 in Section 2.
* (Theoretical contribution) Uniqueness & existence of the solution of the lower-level problem (Proposition 1), Lipchitz continuity of the solution map (Proposition 2), convergence to a stationary point (Theorem 1), and the universal approximation (Theorem 2 & Corollary 1) are proved, ensuring theoretical soundness of the proposed method.
* (Empirical contribution) The proposed method outperforms baselines.
- (Reproducibility) Error bars are provided.

**Weaknesses:**

- Although the theoretical convergence is proved, the proposed method requires a bilevel optimization, which may cause training instability, necessitating warm-start training outlined in Section 3 (please correct me if I missed singing); that said, pretraining or warm-start is often required when training PINN's variants.
- Currently, the code is not available, although it will be released upon acceptance (line 353). Could the authors provide the code? Just a snippet is acceptable.


## Review summary

The paper is well-written and easy to follow. The contribution of the proposed method is clear. I am currently inclined to recommend its acceptance.

**Questions:**

- Can the proposition 1 be extended to nonlinear problems?

---

> ### Author Response · Authors · 2025-11-20
> **W1, W2**
>
> We appreciate the positive assessment and the reviewer's recognition of the clarity and contribution of our work. We address the questions regarding training stability, code availability and extension for proposition 1 below.
>
> ## **W1 Pretraining or warm-start is often required when training PINN's variants.**
> We would like to clarify that pre-training is not a strict requirement for IFeF-PINN in general, but rather an optional strategy that can accelerate convergence, whose necessity is problem-dependent and influenced by the structure of the governing PDE. Since the lower-level problem for linear PDEs achieves a global minimizer based on the current basis functions (as proven in Proposition 1), providing a good initial basis through pre-training can guide the convergence direction and speed up the training process. Meanwhile, it only requires training vanilla PINN for a few hundred epochs to obtain reasonable initial network parameters for the hidden layers, which does not impose significant additional computational cost.
>
> For homogeneous PDEs (with no source term), pre-training is typically necessary to avoid convergence to the trivial zero solution. Standard initialization schemes (Xavier) initialize weights such that the network output is close to zero. Therefore, for homogeneous PDEs, the optimal coefficients $\theta^*$ that minimize the lower-level problem are also near zero. Without using pre-training to obtain the initial features, IFeF-PINN can get trapped in the trivial solution $u \equiv 0$.
>
> For non-homogeneous PDEs like the Helmholtz equation, pre-training is generally not required. The method converges reliably from random initialization because the source term prevents convergence to the trivial solution. Even when pre-training is used for these cases, it serves primarily to accelerate convergence rather than being a necessity for convergence.
>
> ## **W2 Currently, the code is not available, although it will be released upon acceptance (line 353). Could the authors provide the code? Just a snippet is acceptable.**
>
>
> Here is our RFF mapping code.
> ```python
> class RandomFourierFeaturesNN(nn.Module):
>     def __init__(self, input_dim, num_features, sigma=1.0):
>         super(RandomFourierFeaturesNN, self).__init__()
>         self.input_dim = input_dim      # hidden layer input dim
>         self.num_features = num_features # num of random features
>         self.sigma = sigma
>         # sample W from N(0, sigma^2)
>         W = torch.randn(input_dim, num_features, device=device) * sigma # (d, K)
>         self.register_buffer('W', W)
>
>     def forward(self, model, x, y):
>         features = model.get_hidden_output(x, y)  # (N, d)
>         projections = features @ self.W
>         normalization = math.sqrt(2.0 / self.num_features)
>         Phi = normalization * torch.cat([torch.cos(projections), torch.sin(projections)], dim=1)  # (N, 2*num_features)
>         return Phi, projections  # (N, 2*num_features), (N, num_features)
> ```
>
> Here is the code for IFeF-PINN training process.
> ```python
> for epoch in range(n_epochs):
>
>     # --- Feature Extraction ---
>     # --- Phi represents the RFF extended basis, Psi represents the derivatives of Phi under physics residue term ---
>     optimizer.zero_grad()
>     Phi_rff_primal = rff.forward(model, x_data, y_data)[0]
>     Psi_rff_primal = compute_full_RFF_derivatives_double_batched(
>         model, rff, x_phys, y_phys, k, m_chunk_size=25000, k_chunk_size=5000)[-1]
>
>     # --- Lower-level update (Convex QP solver) ---
>     Q_rff = compute_Q(Phi_rff_primal, Psi_rff_primal,
>                       physics_weight_dual, rgl_dual, N, M).detach()
>     c_rff = (- (2/N) * Phi_rff_primal.mT @ u_data
>              - (2*physics_weight_dual/M) * Psi_rff_primal.mT @ q_value).detach()
>
>     with torch.no_grad():
>         L = torch.cholesky(Q_rff)
>         theta_star = torch.cholesky_solve(-c_rff.unsqueeze(1), L).squeeze()
>
>     # --- Upper-level update (Feature learning) ---
>     data_loss = loss_fn(Phi_rff_primal @ theta_star.unsqueeze(1), u_data)
>     phys_loss = torch.mean((Psi_rff_primal @ theta_star.unsqueeze(1) - q_value)**2)
>     loss = data_loss + physics_weight * phys_loss
>
>     # --- Can add Primal-Dual or other training approaches here ---
>
>     loss.backward()
>     optimizer.step()
> ```
> We are willing to provide further details or implementation code for any functions mentioned upon if needed.

---

> ### Author Response · Authors · 2025-11-20
> **Q1**
>
> ## **Q1. Can the proposition 1 be extended to nonlinear problems?**
>
> While Proposition 1 is stated for linear problems where the lower-level problem has a quadratic form with a closed-form global optimum, the framework can indeed be extended to nonlinear problems with appropriate modifications to ensure the convergence of the algorithm.
>
> IFeF-PINN was initially inspired by the QP formulation for linear PDEs. For nonlinear problems, although the convexity of the lower-level problem cannot be guaranteed, we can still find a strict local minimum for the lower-level problem when the Second-Order Sufficient Condition (SOSC) is satisfied (the gradient vanishes and the Hessian matrix is positive definite at the local minimum). Under SOSC, the local minimizer $\theta^{\star}(\omega)$ still has Lipschitz continuity and differentiability properties in a neighborhood of $\omega$, similar to the global minimum in the convex case.
> Consequently, we can follow the same proof pathway as in Appendix C: the solution map $\theta^{\star}(\omega)$ remains locally Lipschitz continuous (Proposition 2 generalized), the hypergradient remains L-smooth (Assumption 2 holds locally), and therefore Theorem 1 on convergence to a stationary point still applies. In practice, for the nonlinear viscous Burgers' equation in our experiments (Section 6.1), we use L-BFGS to solve the lower-level problem to a local minimum, and the algorithm demonstrates stable convergence and comparable approximation performance, empirically validating this extension.
>
> Following the reviewer's comment, we will revise the paper to emphasize that the main contribution and focus are on linear PDEs where the global minimum for lower-level problem is guaranteed (Proposition 1). The extension to nonlinear problems will also be discussed in a subsection, including the extension of proposition 1 to clarify the conditions under which local convergence can be established, with the Burgers' equation serving as a validation for the generality of IFeF-PINN in the experimental section.

---

> > ### Comment · Reviewer_JZcR · 2025-11-26
> >
> > I appreciate the authors for their detailed response, which have resolved my concerns and questions as follows:
> >
> > - Training stability:
> >   - "pre-training is not a strict requirement for IFeF-PINN in general, but rather an optional strategy that can accelerate convergence, whose necessity is problem-dependent and influenced by the structure of the governing PDE"
> >   - Problem-dependent properties are described.
> >
> > - Code availability:
> >   - A snippet is provided.
> >
> > - Extension for proposition 1
> >   - "While Proposition 1 is stated for linear problems where the lower-level problem has a quadratic form with a closed-form global optimum, the framework can indeed be extended to nonlinear problems with appropriate modifications to ensure the convergence of the algorithm."
> >   - Convergence conditions to a local minimum are discussed.
> >
> > I have also read other reviews and found that the authors' responses have adequately addressed the issues raised by the reviewers.
> > Thus, I raised my score to 8.

---

> > > ### Author Response · Authors · 2025-11-26
> > >
> > > We sincerely thank you for your thorough review and for raising the score! We are glad our responses have addressed your concerns. In the revised manuscript, we will explain the pre-training strategy in detail and extend Proposition 1 for nonlinear problems in the Appendix. Thank you again for your valuable feedback and support!

---

### Official Review · Reviewer_Coic · 2025-11-01

**Soundness:** 2
**Presentation:** 2
**Contribution:** 2
**Rating:** 4
**Confidence:** 4

**Summary:**

The paper addresses the challenge of spectral bias in physics-informed neural networks (PINNs) by introducing an iterative training framework that utilizes random Fourier features (RFFs). The approach is based on two types of bases, nominal and extended, and applies a regression method on the extended basis coefficients to learn the PDE solution. The method is evaluated on canonical PDE problems. The paper claims that it is potentially generically applicable to different neural PDE solvers. However, that is not shown in the current version. Some aspects of the paper are unclear, for instance, the current write-up, particularly the detailed construction of the basis, and the motivation for the chosen network layer for RFF extension. Also, the empirical demonstration on nonlinear and highly oscillatory PDEs is a limitation.

**Strengths:**

1. The paper addresses a well-known and important problem of spectral bias in PINNs.

2. The proposed approach is conceptually general and can be integrated with various PINN frameworks.

3. Provides theoretical convergence analysis and demonstrates improvements on canonical problems.

**Weaknesses:**

1. The paper is unclear in how the nominal and extended bases are created or selected, and lacks an explanation for the motivation behind modifying the last layer to improve spectral properties. The presentation of the methodology could be improved.

2. The discussion of the bi-level optimization framework is limited, and different possible formulations and the rationale for the chosen structure are not justified.

3. Computational aspects, such as training cost, convergence behavior, and initialization of the bi-level optimization, are not adequately discussed and compared with prior works.

4. The numerical experiments primarily focus on linear PDEs; the method’s effectiveness for nonlinear oscillatory problems is not demonstrated.

5. A comparison with related approaches, such as [1] and [2], is lacking, and no clear discussion is provided on computational trade-offs or advantages over these methods.

[1] De Ryck, Tim, et al. "An operator preconditioning perspective on training in physics-informed machine learning." The Twelfth International Conference on Learning Representations.

[2] Moseley, Ben, Andrew Markham, and Tarje Nissen-Meyer. "Finite basis physics-informed neural networks (FBPINNs): a scalable domain decomposition approach for solving differential equations." Advances in Computational Mathematics 49.4 (2023): 62.

**Questions:**

1. Could the authors clarify in detail how the nominal and extended bases are constructed and why the RFF extension is applied specifically at the last layer?

2. What are the computational costs and convergence characteristics of the proposed method compared to [1] and [2]? How intensive is the iterative training in practice?

3. How is the initialization of both upper- and lower-level optimizations handled, and how sensitive is the performance to this initialization?

4. Can the authors include results on nonlinear PDEs (e.g., the KdV or Kuramoto–Sivashinsky equations) to demonstrate the method's generality beyond linear problems?

5. Would an ablation on hyperparameter choices (especially for the physics-informed Gaussians and the proposed method) help in understanding performance sensitivity?

6. If the approach primarily targets linear problems, could the authors explicitly position it in relation to [1] and [2], and clarify this limitation in the discussion?

Minor comment:

The formula for computing the relative L2 error is not stated in the paper.

---

> ### Author Response · Authors · 2025-11-20
> **minor comment, W1, Q1**
>
> We appreciate the detailed and constructive feedback regarding the IFeF-PINN structure, computational costs, initialization strategies, and comparisons with related methods are highly valuable, and we address each point in detail below.
>
> ## **Minor comment**
>
> The Relative $L^2$ error formula is given by: $\frac{\lVert u_\text{pred} - u_\text{real} \rVert_2}{\lVert u_\text{real} \rVert_2}$.
>
> ## **W1&Q1. Could the authors clarify in detail how the nominal and extended bases are constructed and why the RFF extension is applied specifically at the last layer?**
>
> We acknowledge that our presentation of the basis construction was not sufficiently clear. We provide a detailed clarification below and will revise the methodology section accordingly.
>
> We define the nominal basis $h_\omega (x)$ as the output of the last hidden layer of a multi-layer perceptron (MLP). Specifically, for an MLP with $L$ hidden layers:
> $$h_\omega(x) = \sigma_L(W_L \sigma_{L-1}(\cdots \sigma_1(W_1 x + b_1) \cdots) + b_L) \in \mathbb{R}^{p}$$
> where $\omega = \[W_1, b_1, \ldots, W_L, b_L\]$ are the learnable parameters, $\sigma$ are activation functions, and $p$ is the dimension of the last hidden layer. This basis of $p$ functions is denoted as the nominal basis. The extended basis is the RFF map of the nominal basis, which is defined as $\psi_D(x) = \gamma_D\left(h_{\omega}(x)\right) = \frac{1}{\sqrt{D}} \begin{bmatrix} \cos(2\pi B_D h_\omega(x)), \\
>             \sin(2\pi B_D h_\omega(x)) \end{bmatrix}$, where $B_D \in R^{D\times p}$ is a constant matrix with entries sampled i.i.d. from $\mathcal{N}(0,\sigma^2)$.
> This transformation maps the $p$-dimensional nominal basis into a $2D$-dimensional feature space, where $D$ can be chosen arbitrarily large to improve expressiveness.
>
> To demonstrate why the RFF extension is applied specifically at the last layer and the improvement for the spectral properties, we provide two complementary justifications.
>
> The primary reason for using RFF mapping in the last hidden layer but not in the input layer derives from the QP formulation of the algorithm for linear PDEs. We want to approximate the PDE solutions using basis functions as $u_{\omega,\theta}(x) = \psi_D(x)^\top \theta$. Using the RFF mapping, we can create an arbitrary number of basis functions and improve the expressiveness of the basis functions. If we apply the RFF in the input layer, then the number of basis functions is limited by the number of neurons in the last hidden layer.
>
> Our theoretical analysis also motivates the structure of our architecture, showing that applying the RFF mapping at the last layer improves spectral properties.
> By Theorem 3 in the Appendix and the proof derived in [1], we show that $k_h(x, x') := k(h(x), h(x')) \approx \psi^\top_D(x) \psi_D(x)$ is a universal kernel and is also a shift-invariant kernel under $h_{\omega}$. Consider the loss function of the lower-level problem
>
> \begin{equation*}
>     \mathfrak{L}(u_{\omega,\theta})
>     = \frac{1}{N_u} \bigl\| B_u(\omega) \theta - G_u \bigr\|^2
>     + \frac{\lambda}{N_f}\bigl\| R_f(\omega) \theta - F_f \bigr\|^2 + \gamma \| \theta \|^2.
> \end{equation*}
> Then we have the normal equation for $\theta$ as
> \begin{equation*}
>     (\frac{1}{N_u} B_u^\top B_u + \frac{\lambda}{N_f} R_f\top R_f + \gamma I)\theta = \frac{1}{N_u} B_u^\top G_u + \frac{\lambda}{N_f} R_f\top F_f
> \end{equation*}
> We can rewrite the lower-level problem using kernel representation to convert the lower-level problem to a kernel ridge regression problem as
> \begin{equation*}
> \Psi \coloneqq \begin{pmatrix} \sqrt{\frac{1}{N_u}}B_u, \\   \sqrt{\frac{\lambda}{N_f}}R_f \end{pmatrix}, \quad
> y \coloneqq \begin{pmatrix} \sqrt{\frac{1}{N_u}}G_u, \\   \sqrt{\frac{\lambda}{N_f}}F_f \end{pmatrix},
> \end{equation*}
> This is precisely the dual form of kernel ridge regression with a positive-definite Gram matrix $K = \Psi \Psi^T \in \mathbb{R}^{(N_u+N_f)\times (N_u+N_f)}$, with the eigenvalue and spectral properties of the kernel directly related to $k_h$. Then using the similar proof path as shown in [1], we can tune $\sigma$ to control the frequency spectrum of the kernel, thereby mitigating spectral bias. This kernel perspective justifies the IFeF-PINN structure: by decoupling the problem into two stages and explicitly employing RFF mapping, we construct a universal shift-invariant kernel $k_h$ whose spectral properties can be controlled through $\sigma$.
>
> [1] Tancik, Matthew, Pratul Srinivasan, Ben Mildenhall, Sara Fridovich-Keil, Nithin Raghavan, Utkarsh Singhal, Ravi Ramamoorthi, Jonathan Barron, and Ren Ng.
> "Fourier features let networks learn high-frequency functions in low-dimensional domains."
> Advances in Neural Information Processing Systems 33 (2020): 7537–7547.

---

> ### Author Response · Authors · 2025-11-20
> **W2**
>
> ## **W2. The discussion of the bi-level optimization framework is limited, and different possible formulations and the rationale for the chosen structure are not justified.**
>
> We provide a comprehensive explanation from three perspectives: (1) natural problem structure, (2) theoretical advantages, and (3) computational considerations.
>
> The initial motivation of IFeF-PINN is to leverage basis function methods for approximating PDE solutions. This approach naturally decomposes into two distinct tasks: Selecting basis functions and regression for the basis coefficients $\theta$. More specifically, when approximating solutions of linear PDEs using basis functions, substituting the basis representation into the standard PINN loss function yields a quadratic programming (QP) problem with respect to the coefficients $\theta$.
> For such a hierarchical structure, bi-level optimization is a direct and principled choice that explicitly separates these two coupled sub-problems.
>
> Beyond the natural problem structure, the bi-level formulation provides strong theoretical benefits. As we prove in Theorem 3.1, the optimal solution $\theta^\star(\omega)$ is Lipschitz continuous with respect to $\omega$. Leveraging the Lipschitz property, we establish convergence guarantees for the entire IFeF-PINN algorithm (see Theorem 3.2). These theoretical guarantees strengthen our confidence in choosing the bi-level structure.
>
> We also want to clarify why we chose alternating optimization (fixing $\omega$ and $\theta$ alternatively) instead of doing more standard bi-level optimization. The key bottleneck is heavy computation cost for the hyper-gradient in Appendix C.1
>
>
>
> \begin{equation*}
>    \nabla_{\omega} \mathfrak{L}_{upper}(\omega) = \\frac{\partial \mathfrak{L}(\omega)} {\partial \omega} - \left(
> \\frac{\partial (Q(\omega)\theta^\star)}{\partial \omega^\top} - \\frac{\partial c(\omega)}{\partial \omega^\top} \right)^\top {Q}(\omega)^{-1} \frac{\partial  \mathfrak{L}}{\partial \theta}.
> \end{equation*}
>
> In the second term, the hypergradient involves prohibitively expensive matrix operations. Both $Q(\omega)$ and $c(\omega)$ depend on the RFF mapped features $\psi_D(x)$, which extend the last hidden layer of the network. This requires a second-order backpropagation over the data and physics points. Moreover, computing $Q(\omega)^{-1}$ in high dimensions has large memory costs, and the subsequent high-order matrix multiplications become infeasible when the number of RFF features is large. Therefore, we choose to use the iterative optimization, which converges faster and has a lower memory usage in practice.

---

> ### Author Response · Authors · 2025-11-20
> **W3, W5, Q2, Q6**
>
> ## **W3, W5, Q2, Q6. Computational costs and convergence characteristics of the proposed method, with comparison with [1] and [2]**
>
> For computational aspects comparison, we record the computational cost and memory usage for IFeF-PINN, Vanilla PINN and the SOTA baseline in our paper, which is the PIG method proposed by [3].
>
> ### **IFeF-PINN**
>
> | Problem | Per upper-level (s) | Per lower-level (s) | Training time (s) | Memory (GB) |
> |--------|----------------------|----------------------|-------------------|-------------|
> | 2D Helmholtz ($a_1=1, a_2=4$) | 0.015 | 0.003 | 448 | 1.41 |
> | 2D Helmholtz ($a_1=a_2=100$)  | 0.154 | 0.042 | 1960 | 18.5 |
> | 1D Convection ($\beta=50$)    | 0.024 | 0.003 | 108 | 4.80 |
> | 1D Convection ($\beta=200$)   | 0.051 | 0.010 | 610 | 5.85 |
> | Convection–Diffusion ($k_{\text{low}}=4\pi$, $k_{\text{high}}=60\pi$) | 0.052 | 0.003 | 110 | 5.26 |
>
> ### **Vanilla**
>
> | Problem | Per epoch (s) | Training time (s) | Memory (GB) |
> |--------|----------------|-------------------|-------------|
> | 2D Helmholtz ($a_1=1, a_2=4$) | 0.003 | 116 | 0.2 |
> | 2D Helmholtz ($a_1=a_2=100$)  | 0.006 | -   | 0.55 |
> | 1D Convection ($\beta=50$)    | 0.002 | 18  | 0.10 |
> | 1D Convection ($\beta=200$)   | 0.002 | -   | 0.10 |
> | Convection–Diffusion ($k_{\text{low}}=4\pi$, $k_{\text{high}}=60\pi$) | 0.003 | 27 | 0.19 |
>
> ### **PIG**
>
> | Problem | Per epoch (s) | Training time (s) | Memory (GB) |
> |--------|----------------|-------------------|-------------|
> | 2D Helmholtz ($a_1=1, a_2=4$) | 0.62 | 248 | 6.8 |
> | 2D Helmholtz ($a_1=a_2=100$)  | 1.70 | -   | 20.6 |
> | 1D Convection ($\beta=50$)    | 1.13 | 565 | 5.8 |
> | 1D Convection ($\beta=200$)   | 1.63 | -   | 14.7 |
> | Convection–Diffusion ($k_{\text{low}}=4\pi$, $k_{\text{high}}=60\pi$) | 0.85 | 43 | 9.8 |
>
> A dash '-' denotes that the method failed to achieve a meaningful approximation for the corresponding equation and is therefore excluded from the total training time.
>
> From the results, we observe that Vanilla PINN, which relies only on MLP training without high-dimensional feature mappings, achieves the fastest per-epoch training speed and the shortest training time. However, it also has the worst fitting performance.
>
> Consider the PIG algorithm, their default optimizer is L-BFGS, bring the longest per-epoch training time. However, it requires the least number of epochs to converge, with only around $50$ epochs on the multi-scale Convection-Diffusion equation (though it fails to capture the high-frequency components of the solution). However, because PIG evaluates and differentiates a large number of learnable Gaussian bases for collocation points, it requires substantial memory usage despite using only a lightweight MLP, as the feature learning and embedding process is memory-intensive. In contrast, our IFeF-PINN algorithm demonstrates lower memory usage compared to PIG. While computation time is higher in some cases, it remains acceptable.
>
> Additionally, we observe from the training process visualization that IFeF-PINN typically achieves the error level for most baselines within relatively few epochs and short training time, with remaining training focused on refining small details for the exact solution at already low error levels. We will include relative $L^2$ error vs. training time curves for some benchmarks in the revision to illustrate this rapid initial convergence of IFeF-PINN.
>
>
>
>
>
> [1] De Ryck, Tim, et al. "An operator preconditioning perspective on training in physics-informed machine learning." The Twelfth International Conference on Learning Representations.
>
> [2] Moseley, Ben, Andrew Markham, and Tarje Nissen-Meyer. "Finite basis physics-informed neural networks (FBPINNs): a scalable domain decomposition approach for solving differential equations." Advances in Computational Mathematics 49.4 (2023): 62.
>
> [3] Kang, Namgyu, Jaemin Oh, Youngjoon Hong, and Eunbyung Park. "PIG: Physics-informed Gaussians as adaptive parametric mesh representations." arXiv preprint arXiv:2412.05994, 2024.

---

> > ### Author Response · Authors · 2025-11-20
> > **Discussion with [1] [2]**
> >
> > We thank the reviewer for bringing these relevant contributions to our attention. We would like to first clarify that our paper already includes comparisons with several representative methods for solving PDEs. In Table 1, we provide a summary of representative PDE solution methods, highlighting their application domains, key ideas, high-frequency handling capabilities, limitations, and optimality properties. Furthermore, we present numerical experiments comparing IFeF-PINN against four baseline methods: Vanilla PINN, PINNsformer, NTK, and PIG. These experiments cover multiple benchmark PDEs, demonstrating that IFeF-PINN achieves superior accuracy under various scenarios.
> >
> > Regarding the reviewer's concern about computational costs, convergence characteristics, and memory usage, we acknowledge that these aspects were not sufficiently considered in the original paper. We have therefore added a comprehensive analysis comparing IFeF-PINN with Vanilla PINN and PIG in terms of total training time, runtime per epoch, and memory usage.
> >
> > [1] addresses PINNs training difficulties from operator conditioning. They claim that the Hermitian operator of the PDE is often ill-conditioned, leading to a very large condition number of the corresponding matrix $A$ in Equation 2.6 related to PINNs formulation. This condition number directly dominates the convergence speed of gradient descent as $\mathcal{O}(\kappa(A) \ln(1/\epsilon))$ iterations to achieve error $\epsilon$. Therefore, they propose preconditioning strategies that rescale the matrix $A$ to improve its conditioning and accelerate convergence. However, their approach also primarily focuses on linear PDEs and requires properties of the Hermitian operator of the PDE to construct effective preconditioners. For complex PDEs, even linear ones, designing appropriate preconditioning can be challenging and may yield poor results if the operator structure is not well understood. In contrast, while IFeF-PINN also primarily targets linear PDEs, it provides a more universal framework. For any linear PDE, we can directly formulate the lower-level QP problem without requiring an understanding of the PDE operator.
> >
> > In our paper, while Theorem 1 establishes convergence to a stationary point, we do not provide an explicit convergence speed analysis. However, we might take a similar perspective from the Hermitian operator of the PDE. In our IFeF-PINN framework,  after applying the differential operator $\mathfrak{F}$ to the extended RFF basis $\psi_D$, we can formulate our lower-level $Q$ matrix. Taking the condition number perspective, there might be a relation between $\kappa(Q)$ and the convergence speed of IFeF-PINN and the Hermitian operator of the PDE.  Moreover, our approach can be viewed as implicitly preconditioning through basis extension rather than explicit matrix rescaling.
> >
> > FBPINNs has some similarities with IFeF-PINN, both approximate PDE using a basis function and focus on the multi-scale and spectral bias in PINNs. FBPINNs is inspired by finite element methods by decomposing the domain into multiple overlapping subdomains, training a small neural network in each subdomain to fit the local solution, and accumulating the global solution from these local basis functions. However, one limitation of FBPINNs is that all experiments in their paper rely on the hard constraint ansatz, which fixes the expression of the function learned by each subdomain. Therefore, for new PDEs or complex boundary conditions, the design for new ansatz formulations is required. Without such ansatz, the method falls back to the standard PINNs formulation, reintroducing the challenges of boundary loss weight tuning, competing residual terms, and gradient pathologies in spectral bias cases. In contrast, IFeF-PINN mitigates spectral bias with a universal basis function formulation, using a universal RFF mapping to construct basis functions as $\psi_D = \gamma_D(h_\omega)$. As demonstrated in our response to Q1, this RFF extension formulates a shift-invariant and universal kernel whose spectral properties can be controlled through $\sigma$.
> >
> > Regarding the training time of FBPINNs, the author claims in Section 6 that "For the single-threaded implementation of our parallel training algorithm used here, the FBPINNs are typically 2 to 10 times slower to train than their corresponding PINNs, despite the FBPINNs being more data-efficient." From the comparison of training time with  IFeF-PINN and Vanilla PINN, we observe that IFeF-PINN's training time is approximately 5 times that of Vanilla PINN, suggesting that IFeF-PINN and FBPINNs may have comparable training time.

---

> ### Author Response · Authors · 2025-11-20
> **Q3, W4&Q4**
>
> ## **Q3. How is the initialization of both upper- and lower-level optimizations handled, and how sensitive is the performance to this initialization?**
>
> Initialization Strategy: the initialization requires the pre-training of standard PINNs for some epochs to get the initial network parameters in the hidden layer for the two-stage iterative process. Whether this is a necessary step is problem-dependent and influenced by the structure of the governing PDE. For homogeneous PDEs (with no source term), pre-training is typically necessary to avoid convergence to the trivial zero solution. Standard initialization schemes (Xavier) initialize weights such that the network output is close to zero. Therefore, for homogeneous PDEs, the optimal coefficients $\theta^*$ that minimize the lower-level problem are also near zero. Without using pre-training to obtain the initial features, IFeF-PINN can get trapped in the trivial solution $u \equiv 0$. For non-homogeneous PDEs like the Helmholtz equation, pre-training is generally not required. The method converges reliably from random initialization because the source term prevents convergence to the trivial solution.
>
> ## **W4&Q4. Can the authors include results on nonlinear PDEs (e.g., the KdV or Kuramoto–Sivashinsky equations) to demonstrate the method's generality beyond linear problems?**
>
> We acknowledge that IFeF-PINN primarily focuses on linear PDEs, where the lower-level problem is a QP problem with a closed-form global optimizer. However, the algorithm also demonstrates generality to certain nonlinear problems with modified optimization strategies. For nonlinear PDEs, where the lower-level problem becomes nonconvex, we consider two iterative approaches: (1) alternating gradient descent with small step sizes for both the upper- and lower-level problems, and (2) finding a local minimum of the lower-level problem periodically during training epochs.
>  We will make this statement clearer and clarify that the main focus of the approach is linear PDEs, while nonlinear PDEs will be discussed in a remark and still used in the benchmark.
>
> We compare the approximation result of Korteweg–De Vries(KdV) equation for IFeF-PINN and Vanilla PINN. We consider the KdV equation with solution:
>
> \begin{equation*}
>  \frac{\partial u}{\partial t} + 6 u \frac{\partial u}{\partial x} +  \frac{\partial^3 u}{\partial x^3} = 0, \quad (t, x) \in [0,1]\times[-4,4],
> \end{equation*}
> \begin{equation*}
>  u(t, x) = 2\operatorname{sech}^2(x - vt), \quad (t, x) \in [0,1]\times[-4,4] .
> \end{equation*}
>
> We use a multi-layer perceptron (MLP) with $4$ hidden layers with $64$ neurons for each layer. We use a fixed dataset with $41$ data points uniformly selected on the boundary and initial condition, and physics points uniformly selected from a $41 \times 41$ grid in the interior of the domain. Therefore the dataset consists of $123$ data points and $1681$ physics points. The following table presents the mean and standard deviation of the relative $L^2$ errors for Vanilla PINN and IFeF-PINN, with the hyperparameter settings for IFeF-PINN in this experiment shown below.
> | Problem       | Vanilla                       | IFeF-PINN                     |
> |---------------|-------------------------------|-------------------------------|
> | KdV equation  | $7.95\times10^{-3}(1.1\times10^{-3})$ | $1.64\times10^{-3}(1.9\times10^{-4})$ |
>
> The results on the KdV equation, together with the viscous Burgers' equation presented in Section 6.1, demonstrate that IFeF-PINN remains effective for nonlinear PDEs. IFeF-PINN achieves approximately 5 times lower error compared to Vanilla PINN on the KdV equation, showcasing its ability to capture the solitary wave solution. While the performance gain is less dramatic than observed for linear high-frequency problems, this is expected due to the loss of convexity in the lower-level problem for nonlinear operators. The lower-level optimization can only guarantee convergence to a local minimum rather than the global optimum achievable in linear cases.

---

> ### Author Response · Authors · 2025-11-20
> **Q5**
>
> ## **Q5. Would an ablation on hyperparameter choices (especially for the physics-informed Gaussians and the proposed method) help in understanding performance sensitivity?**
>
> We acknowledge that while the Spectrum Analysis in Section 6.3 analyzes the impact of different RFF feature dimensions on approximating multi-scale and high-frequency functions, a systematic ablation study covering all key hyperparameters and component necessity (similar to the thorough analysis in Physics-Informed Gaussians) was indeed missing from our original submission.
>
> To address this gap, we conduct extensive ablation studies and report our findings below. We mainly focus on two most important hyperparameters (Feature number $D$ and the bandwidth parameter $\sigma$ in the Gaussian sampling) on two representative benchmarks from our paper (Low-frequency and High-frequency Helmholtz equations).
> ### **Helmholtz ($a_1=1,a_2=4,\sigma=1$)**
> | D                          | 100                | 400                | 800                | 1200               | 3000               | Vanilla            |
> | -------------------------- | ------------------ | ------------------ | ------------------ | ------------------ | ------------------ | ------------------ |
> | **Relative $L^2$ error**   | $5.5\times10^{-4}$ | $2.1\times10^{-4}$ | $3.2\times10^{-4}$ | $5.7\times10^{-4}$ | $4.5\times10^{-4}$ | $2.9\times10^{-2}$ |
> | **Time per iteration (s)** | $0.014$              | $0.015$              | $0.018$              | $0.023 $             | $0.063$              | $0.003$              |
> | **Training time (s)**      | $308$                | $375$                | $448$                |$ 460 $               | $1071$               | $116   $             |
> | **Memory (GB)**            | $0.36$               | $0.69$               | $1.15 $              | $1.62  $             | $3.79 $              | $0.20$               |
>
> ### **Helmholtz ($a_1=1,a_2=4,D=800$)**
> |$\sigma$                | 2                  | 1                  | 0.5                | 0.2                | 0.1                |
> | ------------------------ | ------------------ | ------------------ | ------------------ | ------------------ | ------------------ |
> | **Relative $L^2$ error** | $4.0\times10^{-4}$ | $3.2\times10^{-4}$ | $5.5\times10^{-4}$ | $3.3\times10^{-4}$ |$1.5\times10^{-3}$|
>
> ### **Helmholtz ($a_1=a_2=100,\sigma=1$)**
> | D                          | 800                 | 1200                | 1600                | 2400                | 3000                | Vanilla |
> | -------------------------- | ------------------- | ------------------- | ------------------- | ------------------- | ------------------- | ------- |
> | **Relative $L^2$ error**   | $7.11\times10^{-2}$ | $5.40\times10^{-2}$ | $3.09\times10^{-2}$ |$1.56\times10^{-2}$ | $2.22\times10^{-2}$ | –       |
> | **Time per iteration (s)** | $0.088   $            |$ 0.102  $             | $0.166    $           | $0.196$               | $0.269 $              | $0.006 $  |
> | **Training time (s)**      |$672   $              | $720$                 | $1453$                | $1960 $               |$ 2690 $               | –       |
> | **Memory (GB)**            | $12.2$                | $13.8   $             | $15.4$                |$18.5 $               |$21.0  $              | $0.55$    |
> ### **Helmholtz ($a_1=a_2=100,D=2400$)**
> | $\sigma$                 | 20                 | 10                 | 5                  | 1                   | 0.2                 |
> | ------------------------ | ------------------ | ------------------ | ------------------ | ------------------- | ------------------- |
> | **Relative $L^2$ error** | $4.6\times10^{-3}$ | $3.0\times10^{-3}$ | $5.7\times10^{-3}$ | $1.56\times10^{-2}$ | $1.05\times10^{-1}$ |

---

> > ### Author Response · Authors · 2025-11-20
> > **Analysis for the hyperparameter ablation study**
> >
> > Based on the ablation study results, we obtain significant experimental findings. First, regarding the selection of the number of RFF features $D$, we find that it does not exhibit a linear relationship with the performance. When the number of features is too small, the training time per epoch and memory usage decrease accordingly, but the fitting error also increases due to the lack of features leads to a decrease in the expressivity of the basis functions. However, when the number of RFF features is very large, it may introduce too many ineffective features that negatively impact training or lead to overfitting.
> >
> > Second, regarding the tuning of $\sigma$, we did not have space in the original paper to discuss this hyperparameter in detail. However, as expected from our interpretation of the lower-level problem as kernel ridge regression in W1/Q1, increasing $\sigma$ for high-frequency problems enables RFF features to better capture high-frequency functions. This behavior aligns well with the theoretical and practical analysis in [1] and [2], which demonstrates that higher bandwidth parameters in Fourier features are essential for representing high-frequency components.
> >
> > For the high-frequency Helmholtz equation, increasing $\sigma$ from 1 to 10 reduces the relative $L^2$ error from $0.0156$ to $0.003$, representing an approximately $5$ times improvement. Additionally, we observe from the results that low-frequency problems exhibit lower sensitivity to RFF hyperparameters, with the fitting performance remaining consistently accurate even when $D$ and $\sigma$ vary over a wide range. In contrast, high-frequency problems are more sensitive to RFF hyperparameters, particularly to $\sigma$.
> >
> >
> > [1] Tancik, Matthew, Pratul Srinivasan, Ben Mildenhall, Sara Fridovich-Keil, Nithin Raghavan, Utkarsh Singhal, Ravi Ramamoorthi, Jonathan Barron, and Ren Ng.
> > "Fourier features let networks learn high-frequency functions in low-dimensional domains."
> > Advances in Neural Information Processing Systems 33 (2020): 7537–7547.
> >
> >
> > [2] Wang, Sifan, Hanwen Wang, and Paris Perdikaris.
> > "On the eigenvector bias of Fourier feature networks: From regression to solving multi-scale PDEs with physics-informed neural networks."
> > Computer Methods in Applied Mechanics and Engineering 384 (2021): 113938.

---

> ### Comment · Reviewer_Coic · 2025-11-25
>
> Thanks for the additional experiments and answers in detail. I have raised my score.

---

> > ### Author Response · Authors · 2025-11-26
> >
> > Thank you for your valuable feedback and for raising the score! We greatly appreciate your time and effort and will incorporate some ablation studies, computational cost results, and comparisons with related approaches into the revised manuscript.

---

### Official Review · Reviewer_xJdr · 2025-11-10

**Soundness:** 3
**Presentation:** 3
**Contribution:** 2
**Rating:** 6
**Confidence:** 3

**Summary:**

The paper introduces an iterative training method (IFeF-PINN) that mitigates the spectral bias of Physics-Informed Neural Networks, favoring low-frequency solutions. It separates training into two alternating stages: learning a latent basis through a standard PINN, then extending it with Random Fourier Features (RFF) to capture high-frequency components, followed by a convex regression step. The authors prove convergence for linear PDEs and show that the RFF extension expands the network’s expressive power. Experiments on benchmark PDEs (Helmholtz, convection, Burgers, etc.) demonstrate that IFeF-PINN consistently outperforms existing PINN variants, achieving accurate high-frequency solutions and reduced spectral bias.

**Strengths:**

The paper excels in its clear, well-motivated algorithmic innovation and solid theoretical–empirical balance.

First, it provides a principled bi-level reformulation of PINN training that explicitly decouples feature learning from coefficient regression. This structure not only clarifies optimization dynamics but also leads to provable convexity and convergence for linear PDEs, a strong theoretical contribution.

Second, the integration of Random Fourier Features into the latent space is clever and effective. It directly targets spectral bias instead of relying on heuristics like adaptive weights or resampling.

Third, the paper delivers rigorous ablation and benchmark comparisons (across Helmholtz, convection, Burgers equations) showing that IFeF-PINN achieves one to two orders of magnitude lower relative L2 error than strong baselines, especially under high-frequency and multi-scale regimes where PINNs typically collapse.

Finally, the frequency-domain analysis using FFT provides quantitative evidence that the model indeed learns higher frequencies, reinforcing its claims with interpretable diagnostics.

**Weaknesses:**

A clear weakness of the paper is its limited treatment of nonlinear PDEs, where the proposed convex lower-level formulation breaks down. While the authors acknowledge that the lower regression problem becomes nonconvex in such cases (and suggest it to be promising direction), they did not offer quick remedies or convergence guarantee beyond gradient descent heuristics. This has limited its applicability as many practical PDEs of interest, such as fluid dynamics, reaction–diffusion systems, nonlinear elasticity, are inherently nonlinear. The experiments also mostly focus on linear or mildly nonlinear cases (e.g., Burgers’ equation), leaving open whether the method’s strong performance extends to more challenging nonlinear systems.

**Questions:**

How stable is the iterative bi-level optimization when the Random Fourier Feature dimension D becomes large or when the feature distribution is poorly scaled? Does the alternating update between basis learning and convex regression amplify noise or cause overfitting to spurious high-frequency components?

---

> ### Author Response · Authors · 2025-11-20
> **W1**
>
> We thank the reviewer for the insightful feedback regarding the limitation for nonlinear PDEs and the stability and overfitting of our IFeF-PINN. We address these concerns point-by-point below.
>
> ## **W1. Nonlinear PDEs generality**
>
> We acknowledge that IFeF-PINN primarily focuses on linear PDEs, where the lower-level problem is a QP problem with a closed-form global optimizer. High-frequency linear systems are also widespread and are challenging to solve, arising in areas such as electromagnetic wave propagation, acoustics, and seismic modeling.
> Regarding nonlinear PDEs, the algorithm also generalizes to certain nonlinear problems with modified optimization strategies. For nonlinear PDEs, where the lower-level problem becomes nonconvex, we consider two iterative approaches: (1) alternating gradient descent with small step sizes for both the upper- and lower-level problems, and (2) finding a local minimum of the lower-level problem periodically during training epochs. While the KdV equation also represents a mildly nonlinear case, our experimental results on this problem provide evidence of the method's generality and extensibility to nonlinear systems. To clarify the paper, we will make setup the problem with linear PDEs and add a subsection with the extension to non-linear PDEs and what changes.
>
> We compare the approximation result of Korteweg–De Vries (KdV) equation for IFeF-PINN and Vanilla PINN. We consider the KdV equation with solution:
>
> \begin{equation*}
>  \frac{\partial u}{\partial t} + 6 u \frac{\partial u}{\partial x} +  \frac{\partial^3 u}{\partial x^3} = 0, \quad (t, x) \in [0,1]\times[-4,4]
> \end{equation*}
> \begin{equation*}
>  u(t, x) = 2\\operatorname{sech}^2(x - vt), \quad (t, x) \in [0,1]\times[-4,4] .
> \end{equation*}
>
> We use a multi-layer perceptron (MLP) with $4$ hidden layers with $64$ neurons for each layer. We use a fixed dataset with $41$ data points uniformly selected on the boundary and initial condition, and physics points uniformly selected from a $41 \times 41$ grid in the interior of the domain.
> Therefore the dataset consists of $123$ data points and $1681$ physics points.
> The following table presents the mean and standard deviation of the relative $L^2$ errors for Vanilla PINN and IFeF-PINN, with the hyperparameter settings for IFeF-PINN in this experiment shown below.
> | Problem       | Vanilla                       | IFeF-PINN                     |
> |---------------|-------------------------------|-------------------------------|
> | KdV equation  | $7.95\times10^{-3}(1.1\times10^{-3})$ | $1.64\times10^{-3}(1.9\times10^{-4})$ |
>
> The results on the KdV equation, together with the viscous Burgers' equation presented in Section 6.1, demonstrate that IFeF-PINN remains effective for nonlinear PDEs. IFeF-PINN achieves approximately 5 times lower error compared to Vanilla PINN on the KdV equation, showcasing its ability to capture the solitary wave solution. While the performance gain is less dramatic than observed for linear high-frequency problems, this is expected due to the loss of convexity in the lower-level problem for nonlinear operators. The lower-level optimization can only guarantee convergence to a local minimum rather than the global optimum achievable in linear cases.

---

> ### Author Response · Authors · 2025-11-20
> **Q1**
>
> ## **Q1.How stable is the iterative bi-level optimization when the Random Fourier Feature dimension D becomes large or when the feature distribution is poorly scaled? Does the alternating update between basis learning and convex regression amplify noise or cause overfitting to spurious high-frequency components?**
>
> As shown in Figure 2 and Table 2 of the paper, IFeF-PINN achieves relatively small standard deviations, demonstrating the stability of the method with an appropriate number of RFF features $D$. Moreover, since we employ Tikhonov regularization in the lower-level problem of the two-stage training process as $\theta^{\star}({\omega}) = -(Q({\omega}) + \gamma {I})^{-1} {c}({\omega})$, the numerical stability is further enhanced, preventing some components of $\theta$ from growing too largely and causing overfitting to spurious high-frequency components.
>
> Regarding the feature scaling, we can ensure that the random features mainly cover the frequency spectrum of the target function under appropriate feature number $D$ and bandwidth $\sigma$. The poorly scaled feature may happen when the number of features is too small, while overfitting may occur when the number of features is too large. We conducted an ablation study on both low-frequency and high-frequency Helmholtz equations by varying the feature number $D$ to observe the fitting performance.
>
> ### **Helmholtz** ($a_1 = 1, a_2 = 4$, $\sigma=1$)
>
> | Quantity               | $D=100$           | $D=400$          | $D=800$          | $D=1200$         | $D=3000$         | Vanilla           |
> |------------------------|-------------------|------------------|------------------|------------------|------------------|--------------------|
> | Relative $L^2$ error   | $5.5\times10^{-4}$ | $2.1\times10^{-4}$ | $3.2\times10^{-4}$ | $5.7\times10^{-4}$ | $4.5\times10^{-4}$ | $2.9\times10^{-2}$ |
>
> ### **Helmholtz** ($a_1 = a_2 = 100$, $\sigma=1$)
>
> | Quantity               | $D=800$           | $D=1200$         | $D=1600$         | $D=2400$         | $D=3000$         | Vanilla |
> |------------------------|-------------------|------------------|------------------|------------------|------------------|----------|
> | Relative $L^2$ error   | $7.11\times10^{-2}$ | $5.40\times10^{-2}$ | $3.09\times10^{-2}$ | $1.56\times10^{-2}$ | $2.22\times10^{-2}$ | - |
>
>
>
> From the results, we observe that IFeF-PINN maintains relatively stable and accurate across a wide range of feature numbers $D$ for both low-frequency and high-frequency problems. This demonstrates the robustness of our method: the features are not poorly scaled when the number is small, and the algorithm rarely exhibits overfitting when the number of features is large.
>
>
> This can also be observed from the fitting of the multi-scale equation in Section 6.3 Spectrum Analysis, where we want to fit the $10$ different frequencies:
> \begin{equation*}
> u(x,0)=\sum_{i=1}^{10} A_i \sin( 2 \pi f_i \ x).
> \end{equation*}
> As shown in Figure 5 of the paper, we extracted the top 10 frequency peaks after the Fast Fourier Transform, showing that the frequencies obtained through RFF mapping correspond well with the original function frequencies, and no other spurious high-frequency components are incorrectly fitted.

---

### Meta-Review · Area_Chair_Hi8G · 2026-01-06

**Summary:**

The paper introduces IFeF-PINN, which alleviates PINNs’ spectral bias by injecting Random Fourier Features in the latent-layer representation and optimizing the model via an iterative two-stage procedure, supported by theory for PDE settings and strong accuracy improvements across standard benchmarks.

**Reviewer Concerns:**

Reviewers mainly questioned (i) whether the gains come from the proposed iterative two-stage training rather than a simpler end-to-end variant, (ii) clarity and justification of the RFF design/placement (e.g., last-layer/latent usage) and hyperparameter sensitivity, and (iii) the breadth of evaluation and what the computation cost looks like. The rebuttal addresses these points by clarifying the design rationale, adding ablations (including end-to-end and hyperparameter-related checks), and expanding the experimental evidence (including nonlinear PDE coverage) plus efficiency discussion.

**Reviewer Scores:**

The overall post-rebuttal signal is accept-leaning: the discussion indicates that the authors’ added evidence and clarifications reduce most concerns to positioning rather than fundamental validity, and the final stance across reviews is consistent with maintaining or improving scores toward acceptance.

---

### Decision · Program_Chairs · 2026-01-26

Accept (Poster)